# Diffusion Actor-Critic: Formulating Constrained Policy Iteration as Diffusion Noise Regression for Offline Reinforcement Learning

**Linjiajie Fang**[1]    **Ruoxue Liu**[1]    **Jing Zhang**[1]    **Wenjia Wang**[1,2*]    **Bing-Yi Jing**[3*]
[1] Hong Kong University of Science and Technology
[2] Hong Kong University of Science and Technology (Guangzhou)
[3] Southern University of Science and Technology

## Abstract

In offline reinforcement learning, it is necessary to manage out-of-distribution actions to prevent overestimation of value functions. One class of methods, the policy-regularized method, addresses this problem by constraining the target policy to stay close to the behavior policy. Although several approaches suggest representing the behavior policy as an expressive diffusion model to boost performance, it remains unclear how to regularize the target policy given a diffusion-modeled behavior sampler. In this paper, we propose Diffusion Actor-Critic (DAC) that formulates the Kullback-Leibler (KL) constraint policy iteration as a diffusion noise regression problem, enabling direct representation of target policies as diffusion models. Our approach follows the actor-critic learning paradigm in which we alternatively train a diffusion-modeled target policy and a critic network. The actor training loss includes a soft Q-guidance term from the Q-gradient. The soft Q-guidance is based on the theoretical solution of the KL constraint policy iteration, which prevents the learned policy from taking out-of-distribution actions. We demonstrate that such diffusion-based policy constraint, along with the coupling of the lower confidence bound of the Q-ensemble as value targets, not only preserves the multi-modality of target policies, but also contributes to stable convergence and strong performance in DAC. Our approach is evaluated on D4RL benchmarks and outperforms the state-of-the-art in nearly all environments. [1]

## 1 Introduction

Offline reinforcement learning (RL) aims to learn effective policies from previously collected data, without the need for online interactions with the environment (Levine et al., 2020). It holds promise to implement RL algorithm to real-world applications, where online interactions are risky, expensive, or even impossible. However, learning entirely from offline data brings a new challenge. The prior data, such as human demonstration, are often sub-optimal and cover only a small part of samples compared to the entire state-action space. Learning policies beyond the level of behavior policy demands querying the value function of actions which are often not observed in the dataset. Despite off-policy RL algorithms could be directly applied to offline data, those out-of-distribution (OOD) actions exacerbate the bootstrapping error of value function estimation, typically causing overestimation of action-values and leading to poor performance (Kumar et al., 2019).

To alleviate the problem of overestimation of OOD actions, prior research of policy-regularized algorithm suggests regularizing the learned policy by limiting its deviation from the behavior policy. These methods generally regularize the learned policy by adding a behavior cloning term to the loss function (Fujimoto & Gu, 2021; Wu et al., 2019; Wang et al., 2022) or training a behavior sampler to assist in evaluating the Q-learning target (Kumar et al., 2019; Fujimoto et al., 2019; Wang et al.,

---

*Corresponding authors.
[1]Code is available at https://github.com/Fang-Lin93/DAC.

2022; Chen et al., 2022; Hansen-Estruch et al., 2023). However, due to the intricacy of the behavior distribution, these methods require sufficient representative capacity of models and appropriate regularization schemes to prevent sampling OOD actions and achieve strong performance (Wang et al., 2022).

With the emergence of diffusion models (Sohl-Dickstein et al., 2015; Ho et al., 2020), recent advances on policy-regularized algorithms suggest modeling the behavior policies using high-expressive diffusion models (Janner et al., 2022; Wang et al., 2022; Chen et al., 2022; Hansen-Estruch et al., 2023). However, there are several limitations with the current implementations of diffusion models in offline RL. Some methods use the diffusion model as a behavior sampler for subsequent action generation (Chen et al., 2022; Hansen-Estruch et al., 2023). Those methods require generating lots of action candidates to choose from, which hinders the real-world applications for the slow inference process. Diffusion Q-learning (Wang et al., 2022) trains a biased diffusion model to aid in the estimation of the Q-learning target. Nevertheless, the biased diffusion model no longer prevents from sampling OOD actions (Figure 1 & 2), and the back-propagation of gradients through the denoising process makes the training process time-consuming. Additionally, another drawback of modeling behavior policies as diffusion models is the inability of the diffusion models to explicitly estimate density values. Techniques that rely on access to density functions are not directly applicable given a diffusion-modeled behavior policy (Peng et al., 2019; Nair et al., 2020). Furthermore, it remains unclear how to regularize policies to stay close to a diffusion-modeled behavior that is both theoretically sound and effective in practical performance.

In this paper, we propose Diffusion Actor-Critic (DAC) to address the offline RL problem by training a diffusion-modeled target policy. We focus on the optimization problem of constrained policy iteration (Schulman et al., 2015; Peng et al., 2019; Nair et al., 2020; Chen et al., 2022), where the target policy is trained to maximize the estimated Q-function while fulfilling the KL constraint of the data distribution. We derive that the optimization problem can be formulated as a diffusion noise regression problem, eliminating the need for explicit density estimation of either the behavior policy or the target policy. The resulting noise prediction target involves a soft Q-guidance term that adjusts the Q-gradient guidance according to the noise scales, which distinguishes it from both the guided sampling with return prompts (Janner et al., 2022; Chen et al., 2021) and methods where the Q-gradient is applied to the denoised action samples (Wang et al., 2022). DAC follows the actor-critic learning paradigm, where we alternatively train a diffusion-modeled target policy and an action-value model. During the actor learning step, we train policy model by regressing on a target diffusion noise in a supervised manner. For the critic learning, we employ the lower confidence bound (LCB) of a Q-ensemble to stabilize the estimation of Q-gradients under function approximation error. This approach prevents the detrimental over-pessimistic bias of taking the ensemble minimum as used in the previous research (Fujimoto et al., 2018; Fujimoto & Gu, 2021; Wang et al., 2022). Experiments demonstrate that the LCB target balances the overestimation and underestimation of the value target, leading to improved performance.

In conclusion, our main contributions are:

- Introducing DAC, a new offline RL algorithm that directly generates the target policy using diffusion models. The high-expressiveness of diffusion models is able to capture not only the multi-modality of behavior polices, but also the complexity of target polices as well. Moreover, the training of DAC avoids the back-propagation of gradients through the denoising path, which significantly saves the training time for learning diffusion policies.

- Proposing the soft Q-guidance that analytically solves the KL constraint policy iteration using diffusion models, without the need for explicit density estimation of either the behavior policy or the target policy. The necessity for constraint satisfaction in learning diffusion-modeled target policies is not only crucial for theoretical comprehension but also guarantees that the generated policy refrains from taking OOD actions.

- We demonstrate the effectiveness of DAC on the D4RL benchmarks and observe that it outperforms nearly all prior methods by a significant margin, thereby establishing a new state-of-the-art baseline. Additionally, DAC shows stable convergence and strong performance without the need for online model selection (Wang et al., 2022; Kang et al., 2024), making it more practical for real-world applications.

## 2 PRELIMINARIES

We consider the RL problem formulated as an infinite horizon discounted Markov Decision Process (MDP), which is defined as a tuple $(\mathcal{S}, \mathcal{A}, \mathcal{T}, d_0, r, \gamma)$ (Sutton & Barto, 1999) with state space $\mathcal{S}$, action space $\mathcal{A}$, transition probabilities $\mathcal{T}(\mathbf{s}'|\mathbf{s}, \mathbf{a})$, initial state distribution $\mathbf{s}_0 \sim d_0$, reward function $r(\mathbf{s}, \mathbf{a})$, and discount factor $\gamma \in (0, 1)$. The goal of RL is to train a policy $\pi(\mathbf{a}|\mathbf{s}) : \mathcal{A} \times \mathcal{S} \rightarrow [0, 1]$ that maximizes the expected return: $J(\pi) := \mathbb{E}_{\pi, \mathcal{T}, d_0}[\sum_{t=0}^{\infty} \gamma^t r(\mathbf{s}_t, \mathbf{a}_t)]$. We also define the discounted state visitation distribution $d^\pi(\mathbf{s}) := (1 - \gamma) \sum_{t=0}^{\infty} \gamma^t p_\pi(\mathbf{s}_t = \mathbf{s})$. Then the RL objective $J(\pi)$ has an equivalent form as maximizing the expected per-state-action rewards: $\tilde{J}(\pi) = \mathbb{E}_{\mathbf{s} \sim d^\pi, \mathbf{a} \sim \pi(\cdot|\mathbf{s})}[r(\mathbf{s}, \mathbf{a})]$ (Nachum & Dai, 2020). In offline RL, the agent has only access to a static dataset $\mathcal{D}$, which is collected by a potentially unknown behavior policy $\pi_\beta$, without the permission to fetch new data from the environment.

**Constrained policy iteration.** Let $Q^\pi : \mathcal{S} \times \mathcal{A} \rightarrow \mathbb{R}$ be the Q-function of the policy $\pi$, which is defined by $Q^\pi(\mathbf{s}, \mathbf{a}) = \mathbb{E}_{\pi, \mathcal{T}}[\sum_{t=0}^{\infty} \gamma^t r(\mathbf{s}_t, \mathbf{a}_t)|\mathbf{s}_0 = \mathbf{s}, \mathbf{a}_0 = \mathbf{a}]$. In a standard policy iteration paradigm at iteration $k$, the algorithm iterates between improving the policy $\pi_k$ and estimating the Q-function $Q^{\pi_k}$ via Bellman backups (Sutton & Barto, 1999). Estimating $Q^{\pi_k}$ in the offline setting may request OOD actions that are not observed in the dataset, resulting in an accumulation of bootstrapping errors. To address this issue, off-policy evaluation algorithms (Fujimoto et al., 2019; Kumar et al., 2019; Wu et al., 2019; Schulman et al., 2015; Peng et al., 2019; Nair et al., 2020) propose to explicitly regularize the policy improvement step, leading to the constrained optimization problem:

$$\pi_{k+1} = \arg\max_\pi \mathbb{E}_{\mathbf{s} \sim d^{\pi_k}}[\mathbb{E}_{\mathbf{a} \sim \pi(\cdot|\mathbf{s})} Q^{\pi_k}(\mathbf{s}, \mathbf{a})]$$
$$\text{s.t. } D(\pi, \pi_\beta) \leq \epsilon_b. \tag{1}$$

Commonly used constraints for $D$ are members from $f$-divergence family, such as KL-divergence, $\chi^2$-divergence and total-variation distance (Peng et al., 2019; Nair et al., 2020; Nachum et al., 2019; Nachum & Dai, 2020). In this paper we consider $D$ being the expected state-wise (reverse) KL-divergence: $D(\pi, \pi_\beta) = \mathbb{E}_{\mathbf{s} \sim d^\pi} D_{\text{KL}}(\pi(\cdot|\mathbf{s})\|\pi_\beta(\cdot|\mathbf{s}))$. However, the complicated dependency of $d^{\pi_k}$ on $\pi_k$ makes it difficult to directly solve the KL constraint optimization problem (1) in offline RL. A typical approach for addressing this issue involves substituting the on-policy distribution $d^{\pi_k}$ with the off-policy dataset $\mathcal{D}$ (Peng et al., 2019; Nair et al., 2020), resulting in the surrogate objective:

$$\pi_{k+1} = \arg\max_\pi \mathbb{E}_{\mathbf{s} \sim \mathcal{D}}[\mathbb{E}_{\mathbf{a} \sim \pi(\cdot|\mathbf{s})} Q^{\pi_k}(\mathbf{s}, \mathbf{a})]$$
$$\text{s.t. } \mathbb{E}_{\mathbf{s} \sim \mathcal{D}}[D_{\text{KL}}(\pi(\cdot|\mathbf{s})\|\pi_\beta(\cdot|\mathbf{s}))] \leq \epsilon_b, \tag{2}$$

where $\epsilon_b$ is a pre-defined hyperparameter to control the strength of the constraint.

**Diffusion models.** Diffusion models (Sohl-Dickstein et al., 2015; Ho et al., 2020; Song et al., 2020) are generative models that assumes latent varibles following a Markovian noising and denoising process. The forward noising process $\{\mathbf{x}_{0:T}\}$ gradually adds Gaussian noise to the data $\mathbf{x}_0 \sim p(\mathbf{x}_0)$ with a pre-defined noise schedule $\{\beta_{1:T}\}$:

$$q(\mathbf{x}_{1:T}|\mathbf{x}_0) = \prod_{t=0}^{T} q(\mathbf{x}_t|\mathbf{x}_{t-1}), \ q(\mathbf{x}_t|\mathbf{x}_{t-1}) := \mathcal{N}(\mathbf{x}_t; \sqrt{1 - \beta_t}\mathbf{x}_{t-1}, \beta_t \mathbf{I}). \tag{3}$$

The joint distribution in (3) yields an analytic form of the marginal distribution

$$q_t(\mathbf{x}_t|\mathbf{x}_0) = \mathcal{N}(\mathbf{x}_t; \sqrt{\bar{\alpha}_t}\mathbf{x}_0, (1 - \bar{\alpha}_t)\mathbf{I}) \text{ for all } t \in \{1, ..., T\}, \tag{4}$$

using the notation $\alpha_t := 1 - \beta_t$ and $\bar{\alpha}_t := \prod_{s=1}^{t} \alpha_s$. Given $\mathbf{x}_0$, the noisy sample $\mathbf{x}_t$ can be easily obtained through the re-parameterization trick:

$$\mathbf{x}_t = \sqrt{\bar{\alpha}_t}\mathbf{x}_0 + \sqrt{1 - \bar{\alpha}_t}\boldsymbol{\epsilon}, \ \boldsymbol{\epsilon} \sim \mathcal{N}(\mathbf{0}, \mathbf{I}). \tag{5}$$

DDPMs (Ho et al., 2020) use parameterized models $p_\theta(\mathbf{x}_{t-1}|\mathbf{x}_t) = \mathcal{N}(\mathbf{x}_{t-1}; \boldsymbol{\mu}_\theta(\mathbf{x}_t, t), \boldsymbol{\Sigma}_\theta(\mathbf{x}_t, t))$ to reverse the diffusion process: $p_\theta(\mathbf{x}_{0:T}) = \mathcal{N}(\mathbf{x}_T; \mathbf{0}, \mathbf{I}) \prod_{t=1}^{T} p_\theta(\mathbf{x}_{t-1}|\mathbf{x}_t)$. The practical implementation involves directly predicting the Gaussian noise $\boldsymbol{\epsilon}$ in (5) using a neural network $\boldsymbol{\epsilon}_\theta(x_t, t)$ to minimize the original evidence lower bound loss:

$$\mathcal{L}(\theta) = \mathbb{E}_{\mathbf{x}_0 \sim p(\mathbf{x}_0), t \sim \text{Unif}(1, T), \boldsymbol{\epsilon} \sim \mathcal{N}(\mathbf{0}, \mathbf{I})} \|\boldsymbol{\epsilon} - \boldsymbol{\epsilon}_\theta(\sqrt{\bar{\alpha}_t}\mathbf{x}_0 + \sqrt{1 - \bar{\alpha}_t}\boldsymbol{\epsilon}, t)\|^2. \tag{6}$$

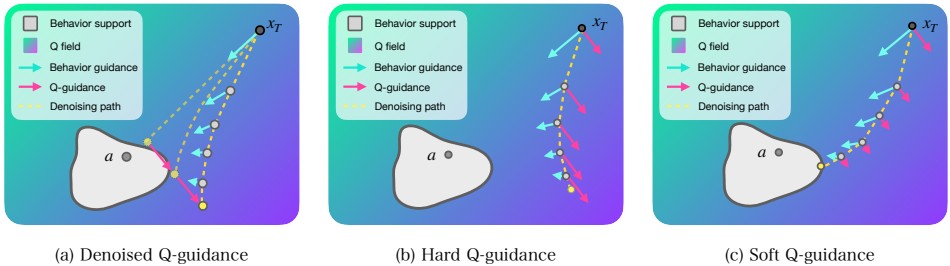

Figure 1: A visual explanation of generating actions from noisy prior $x_T$ using (a) denoised Q-guidance (b) hard Q-guidance and (c) soft Q-guidance. The soft Q-guidance reduces the intensity of Q-guidance during the denoising steps, generating high-reward actions within the behavior support without the need to backpropagate the gradient through the path.

A natural approach to employing diffusion models in behavior cloning involves replacing the noise predictor with a state-conditional model $\epsilon_\theta(\mathbf{x}_t, \mathbf{s}, t)$ that generates actions $\mathbf{x}_0 \in \mathcal{A}$ based on state $\mathbf{s}$.

**Score-based models.** The key idea of score-based generative models (Vincent, 2011; Song & Ermon, 2019; 2020) is to estimate the (Stein) score function, which is defined as the gradient of the log-likelihood $\nabla_\mathbf{x} \log p(\mathbf{x})$. Like diffusion models, score-based models perturb the data with a sequence of Gaussian noise and train a deep neural network $s_\theta(\mathbf{x}_t, t)$ to estimate the score $\nabla_{\mathbf{x}_t} \log p(\mathbf{x}_t)$ for noisy samples $\mathbf{x}_t \sim \mathcal{N}(\mathbf{x}_t; \mathbf{x}_0, \sigma_t^2 \mathbf{I})$ on different noise levels $t = 1, 2, ..., T$. The objective of explicit score matching (Vincent, 2011) is given by:

$$\mathbb{E}_{x_0 \sim p, x_t \sim \mathcal{N}(\mathbf{x}_t; \mathbf{x}_0, \sigma_t^2 \mathbf{I}), t \sim \text{Unif}(1,T)}[\lambda(t)||\nabla_\mathbf{x} \log p(\mathbf{x}) - s_\theta(\mathbf{x}_t, t)||^2], \tag{7}$$

where $\lambda(t) > 0$ is a positive weighting function. Once the estimated score functions have been trained, samples are generated using score-based sampling techniques, such as Langevin dynamics (Song & Ermon, 2019) and stochastic differential equations (Song et al., 2020).

## 3 DIFFUSION ACTOR-CRITIC

In this section, we introduce the Diffusion Actor-Critic (DAC) framework that models the target policy directly as a diffusion model, eliminating the need for density estimation of either the behavior policy or the target policy. Initially, we formulate the KL constraint policy optimization as a diffusion noise regression problem, which yields a soft Q-guidance term for the noise prediction process that enables the learning of the target policy in a supervised manner. Additionally, we introduce Q-ensemble to stabilize the Q-gradient estimation, which utilizes LCB to mitigate the over-pessimistic estimation associated with taking the ensemble minimum in prior research.

### 3.1 LEARNING DIFFUSION POLICY THROUGH SOFT Q-GUIDANCE

The problem of behavior constraint policy iteration (2) has a closed form solution $\pi_{k+1}^*$ (Peng et al., 2019; Nair et al., 2020; Chen et al., 2022) by utilizing the Lagrangian multiplier:

$$\pi_{k+1}^*(\mathbf{a}|\mathbf{s}) = \frac{1}{Z(\mathbf{s})} \pi_\beta(\mathbf{a}|\mathbf{s}) \exp\left(\frac{1}{\eta} Q^{\pi_k}(\mathbf{s}, \mathbf{a})\right), \tag{8}$$

where $\eta > 0$ is a Lagrangian multiplier and $Z(\mathbf{s})$ is a state-conditional partition function. Obtaining the closed form solution of $\pi_{k+1}^*$ directly from (8) is challenging as it requires estimation of the density function of the behavior policy $\pi_\beta$ and the partition function $Z(\mathbf{s})$. Prior methods (Peng et al., 2019; Nair et al., 2020; Chen et al., 2022) suggest addressing this issue by projecting $\pi_{k+1}^*$ onto a parameterized policy $\pi_\theta$ using KL-divergence:

$$\arg\min_\theta \mathbb{E}_{\mathbf{s} \sim \mathcal{D}}[D_{KL}(\pi_{k+1}^*(\cdot|\mathbf{s}) \| \pi_\theta(\cdot|\mathbf{s}))], \tag{9}$$

resulting in the policy update algorithm:

$$\theta_{k+1} = \arg\max_\theta \mathbb{E}_{(\mathbf{s},\mathbf{a}) \sim \mathcal{D}}[\log \pi_\theta(\mathbf{a}|\mathbf{s}) \exp\left(\frac{1}{\eta} Q^{\pi_k}(\mathbf{s}, \mathbf{a})\right)]. \tag{10}$$

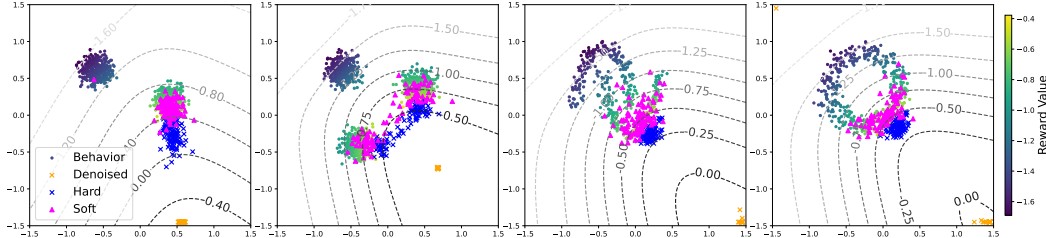

Figure 2: Comparison of generated policies on 2-dimensional bandit using different Q-gradient guidance. We compare soft Q-guidance (magenta) against hard Q-guidance (blue) that eliminates the noise scaling factor and denoised Q-guidance (Wang et al., 2022) (brown) on 2-D bandit examples. The dots are behavior policies, which are colored based on the reward value. The dashed level curves represent the estimated Q-value field. Soft Q-guidance are capable of generating high-reward actions while remaining within the behavior support. We also observe that soft Q-guidance captures the multi-modality of target policies as shown in the second plot. Experimental details can be found in Appendix B.5.

Although (10) eliminates the necessity for estimating the partition function $Z(\mathbf{s})$ and the behavior policy $\pi_\beta$, it needs explicit modeling of the density function of the target policy $\pi_\theta$. Such requirement for $\pi_\theta$ makes it unfeasible to directly use diffusion generative models due to the unavailability of density function estimation. Prior methods employ Gaussian policies to estimate $\pi_\theta$ (Peng et al., 2019; Nair et al., 2020), which limits the expressiveness of the target policy. To address these issues, we rewrite (8) using score functions:

$$\nabla_{\mathbf{a}} \log \pi_{k+1}^*(\mathbf{a}|\mathbf{s}) = \nabla_{\mathbf{a}} \log \pi_\beta(\mathbf{a}|\mathbf{s}) + \frac{1}{\eta}\nabla_{\mathbf{a}} Q^{\pi_k}(\mathbf{s},\mathbf{a}), \ \mathbf{a} \in \mathcal{A}, \tag{11}$$

where the action space $\mathcal{A}$ is usually a compact set in $\mathbb{R}^d$ for $d$-dimensional actions. It seems that the target score function $\nabla_{\mathbf{a}} \log \pi_{k+1}^*(\mathbf{a}|\mathbf{s})$ can be trained by regression on the right-hand-side. However, since $\pi_\beta$ is unknown, we do not have the explicit regression target $\nabla_{\mathbf{a}} \log \pi_\beta(\mathbf{a}|\mathbf{s})$. Drawing inspiration from explicit score matching with finite samples (Vincent, 2011), we smoothly extend the policy functions and the value function defined in $\mathcal{A}$ to the extended action space $\mathbb{R}^d$. Then we consider (11) of the optimal score functions to hold for noisy perturbations of the observation set:

$$\nabla_{\mathbf{x}_t} \log p_t^*(\mathbf{x}_t|\mathbf{s}) = \nabla_{\mathbf{x}_t} \log p_t(\mathbf{x}_t|\mathbf{s}) + \frac{1}{\eta}\nabla_{\mathbf{x}_t} Q^{\pi_k}(\mathbf{s},\mathbf{x}_t), \ \mathbf{x}_t \in \mathbb{R}^d, \tag{12}$$

where $p_t^*(\mathbf{x}_t|\mathbf{s}) = \int q_t(\mathbf{x}_t|\mathbf{a})\pi_{k+1}^*(\mathbf{a}|\mathbf{s})d\mathbf{a}$ and $p_t(\mathbf{x}_t|\mathbf{s}) = \int q_t(\mathbf{x}_t|\mathbf{a})\pi_\beta(\mathbf{a}|\mathbf{s})d\mathbf{a}$ are noise distributions. The noisy perturbation $\mathbf{x}_t \sim q_t(\mathbf{x}_t|\mathbf{a})$ is defined in (4) with $\mathbf{x}_0 = \mathbf{a}$. When the perturbation is small, i.e. $q_t(\mathbf{x}_t|\mathbf{a}) \approx \delta(\mathbf{x}_t - \mathbf{a})$, then $p_t^*(\mathbf{x}_t|\mathbf{s}) \approx \pi_{k+1}^*(\mathbf{a}|\mathbf{s})$ and $p_t(\mathbf{x}_t|\mathbf{s}) \approx \pi_\beta(\mathbf{a}|\mathbf{s})$, which recovers the relationship between score functions within the action space $\mathcal{A}$ as described in (11). Tackling the score function of noise distribution is favorable, since $\nabla_{\mathbf{x}_t} \log p_t^*(\mathbf{x}_t|\mathbf{s})$ itself serves as a means of generating $\pi_{k+1}^*$ using diffusion models, without the need for score-based sampling methods such as Langevin dynamics, as described in the following theorem.

**Theorem 1.** *Let $\boldsymbol{\epsilon}^*(\mathbf{x}_t, \mathbf{s}, t) := -\sqrt{1 - \bar{\alpha}_t}\nabla_{\mathbf{x}_t} \log p_t^*(\mathbf{x}_t|\mathbf{s})$. Then $\boldsymbol{\epsilon}^*(\mathbf{x}_t, \mathbf{s}, t)$ is a Gaussian noise predictor which defines a diffusion model for generating $\pi_{k+1}^*$.*

Although $\boldsymbol{\epsilon}^*(\mathbf{x}_t, \mathbf{s}, t)$ determines the diffusion model that directly generates the target policy, the form of the target noise $\boldsymbol{\epsilon}^*(\mathbf{x}_t, \mathbf{s}, t)$ necessitates the estimation of the noisy score function of the behavior policy $\nabla_{\mathbf{x}_t} \log p_t(\mathbf{x}_t|\mathbf{s})$ by (12), which is typically not accessible. To tackle this problem, we investigate the learning objective when utilizing function approximators. Specifically, we project the target noise $\boldsymbol{\epsilon}^*(\mathbf{x}_t, \mathbf{s}, t)$ onto a parameterized conditional noise model $\boldsymbol{\epsilon}_\theta(\mathbf{x}_t, \mathbf{s}, t)$ via $L^2$-loss, following the standard training objective of diffusion models:

$$\arg\min_\theta \mathbb{E}_{\mathbf{s}\sim\mathcal{D},\mathbf{a}\sim\pi^*,\mathbf{x}_t\sim q_t(\mathbf{x}_t|\mathbf{a}),t}||\boldsymbol{\epsilon}_\theta(\mathbf{x}_t, \mathbf{s}, t) - \boldsymbol{\epsilon}^*(\mathbf{x}_t, \mathbf{s}, t)||^2. \tag{13}$$

To eliminate the need for sampling from the unknown target policy $\pi_{k+1}^*$, we approximate the expectation through $\mathbf{a} \sim \pi_{k+1}^*$ by the behavior data $\mathbf{a} \sim \mathcal{D}$, resulting in the surrogate objective:

$$\arg\min_\theta \mathbb{E}_{(\mathbf{s},\mathbf{a})\sim\mathcal{D},\mathbf{x}_t\sim q_t(\mathbf{x}_t|\mathbf{a}),t}||\boldsymbol{\epsilon}_\theta(\mathbf{x}_t, \mathbf{s}, t) - \boldsymbol{\epsilon}^*(\mathbf{x}_t, \mathbf{s}, t)||^2. \tag{14}$$

---

**Algorithm 1** Diffusion Actor-Critic Training

---

**Require:** offline dataset $\mathcal{D}$, batch size $B$, learning rates $\alpha_\phi$, $\alpha_\theta$, $\alpha_\eta$ and $\alpha_{\text{ema}}$, behavior cloning threshold $b$, pessimism factor $\rho$, initial Lagrangian multiplier $\eta_{\text{init}}$, ensemble size $H$

1: Initialize: diffusion policy $\epsilon_\theta$, target diffusion policy $\epsilon_{\bar{\theta}} = \epsilon_\theta$, Q-networks $Q_{\phi^h}$, target Q-networks $Q_{\bar{\phi}^h} = Q_{\phi^h}$ ($h = 1, 2, ..., H$), Lagrangian multiplier $\eta = \eta_{\text{init}}$
2: **while** training not convergent **do**
3:     Sample a batch of $B$ transitions $\{(\mathbf{s}, \mathbf{a}, r, \mathbf{s}')\} \subset \mathcal{D}$
4:     Sample $\mathbf{a}' = \mathbf{x}_0$ through denoising process using noise predictor $\epsilon_{\bar{\theta}}(\mathbf{x}_t, t, \mathbf{s})$.
5:     **for** $h$ in $\{1, 2, ..., H\}$ **do**
6:         Update $\phi^h \leftarrow \phi^h - \alpha_\phi \nabla_{\phi^h} \mathcal{L}_C(\phi^h)$ (17)                 ▷ Critic learning
7:     **end for**
8:     Sample $\epsilon \sim \mathcal{N}(\mathbf{0}, \mathbf{I})$, $t \sim \text{Unif}(0, T)$ and compute $\mathbf{x}_t = \sqrt{\bar{\alpha}_t}\mathbf{a} + \sqrt{1 - \bar{\alpha}_t}\epsilon$
9:     Estimate Q-gradient $\nabla_{\mathbf{x}_t} Q^{\pi_k}(\mathbf{s}, \mathbf{x}_t)$ using (18)
10:    $\theta \leftarrow \theta - \alpha_\theta \nabla_\theta \mathcal{L}_A(\theta)$ (16)                           ▷ Actor learning
11:    $\eta \leftarrow \eta + \alpha_\eta(||\epsilon_\theta(\mathbf{x}_t, \mathbf{s}, t) - \epsilon||^2 - b)$       ▷ Dual gradient ascent (optional)
12:    $\bar{\theta} \leftarrow (1 - \alpha_{\text{ema}})\bar{\theta} + \alpha_{\text{ema}}\theta$
13:    $\bar{\phi}^h \leftarrow (1 - \alpha_{\text{ema}})\bar{\phi}^h + \alpha_{\text{ema}}\phi^h$         ▷ Update target networks using EMA
14: **end while**

---

Such learning objective has an equivalent form which is easy to optimize.

**Theorem 2.** *Training parameters $\theta$ according to (14) is equivalent to optimize the following objective:*

$$\arg\min_\theta \mathbb{E}_{(\mathbf{s},\mathbf{a})\sim\mathcal{D},\epsilon\sim\mathcal{N}(\mathbf{0},\mathbf{I}),t}||\epsilon_\theta(\mathbf{x}_t, \mathbf{s}, t) - \epsilon + \frac{1}{\eta}\sqrt{1 - \bar{\alpha}_t}\,\nabla_{\mathbf{x}_t} Q^{\pi_k}(\mathbf{s}, \mathbf{x}_t)||^2, \qquad (15)$$

*where $\mathbf{x}_t = \sqrt{\bar{\alpha}_t}\mathbf{a} + \sqrt{1 - \bar{\alpha}_t}\epsilon$.*

The learning objective (15) defines a noise regression problem that approximates the solution of the KL constraint policy iteration (2) within the diffusion model framework, without requiring the estimation of densities for either the behavior policy or the target policy. We refer the last term in the noise target as the *soft Q diffusion guidance* or simply *soft Q-guidance*. Within the soft Q-guidance, the Q-gradient is weighted by the noise scale $\sqrt{1 - \bar{\alpha}_t}$. In a typical diffusion model, the noise scale $\sqrt{1 - \bar{\alpha}_t} \rightarrow 0$ as $t \rightarrow 0$ during the denoising process. This suggests that soft Q-guidance encourages the exploration of high-reward regions in the initial steps of the denoising process, and then gradually fades the guidance strength as the denoising step approaches the final output. In comparison to the hard Q-guidance that eliminates the noise scaling factor or guidance on denoised actions (see Figure 1 for illustrations), soft Q-guidance produces high-fidelity actions that closely resemble the behavior policies, thereby preventing the sampling of out-of-distribution (OOD) actions (Figure 2).

To connect the learning objective (15) with policy-regularized methods, we rearrange the terms in (15) and incorporate constant(s) into $\eta$, resulting in the following actor learning loss:

$$\mathcal{L}_A(\theta) = \mathbb{E}_{(\mathbf{s},\mathbf{a})\sim\mathcal{D},\epsilon,t}\left[\eta||\epsilon_\theta(\mathbf{x}_t, \mathbf{s}, t) - \epsilon||^2 + \sqrt{1 - \bar{\alpha}_t}\,\epsilon_\theta(\mathbf{x}_t, \mathbf{s}, t) \cdot \nabla_{\mathbf{x}_t} Q^{\pi_k}(\mathbf{s}, \mathbf{x}_t)\right], \qquad (16)$$

where the dot ($\cdot$) implies inner product. The Lagrangian multiplier $\eta$ determines the trade-off between the behavior cloning and the policy improvement. As $\eta \rightarrow \infty$, the noise prediction loss (16) reduces to behavior cloning using a parameterized conditional diffusion model, as used in the recent research (Chen et al., 2022; Hansen-Estruch et al., 2023; Wang et al., 2022). The second term involves a inner product between the predicted noise and the Q-gradient, promoting the acquired denoising directions to align with the estimated Q-gradient field. In practical implementation, DAC balances the behavior cloning and policy improvement by controlling $\eta$ to be either fixed or learnable through dual gradient ascent. If $\eta$ is learnable, DAC trains $\eta$ to ensure that $||\epsilon_\theta(\mathbf{x}_t, \mathbf{s}, t) - \epsilon||^2 \leq b$ for a given threshold value $b > 0$.

During critic learning, we approximate $Q^{\pi_k}$ with neural networks. To enhance the stability of estimating the Q-gradient used in (16), we follow the method of using pessimistic Q-ensembles (Wu et al., 2019; Agarwal et al., 2020; Smit et al., 2021; Lee et al., 2021; 2022; An et al., 2021). Specifically, we train an ensemble of $H$ parameterized Q-networks $Q_{\phi^h_k}$ and target Q-networks $Q_{\bar{\phi}^h_k}$, along with lower confidence bound (LCB) as value targets (Ghasemipour et al., 2022), which leads to the

critic learning loss:

$$\mathcal{L}_C(\phi^h) = \mathbb{E}_{(\mathbf{s},\mathbf{a},r,\mathbf{s}')\sim\mathcal{D},\mathbf{a}'\sim\pi_{\theta_k}}\left[r + \gamma Q_{\text{LCB}}(\mathbf{s}',\mathbf{a}') - Q_{\phi^h}(\mathbf{s},\mathbf{a})\right]^2,$$
$$Q_{\text{LCB}}(\mathbf{s}',\mathbf{a}') = \mathbb{E}_h[Q_{\bar{\phi}_k^h}(\mathbf{s}',\mathbf{a}')] - \rho\sqrt{\text{Var}_h[Q_{\bar{\phi}_k^h}(\mathbf{s}',\mathbf{a}')]}, \tag{17}$$

where $\rho \geq 0$ is a hyperparameter that determines the level of pessimism, and $\mathbb{E}_h[\cdot]$ and $\text{Var}_h[\cdot]$ are empirical mean and variance operators over the $H$ ensembles. Once the Q-functions are trained, the Q-gradient in the soft Q-guidance can be estimated by the ensemble's average of target Q-networks:

$$\nabla_{\mathbf{x}_t}Q^{\pi_k}(\mathbf{s},\mathbf{x}_t) \approx \frac{1}{HC}\sum_{h=1}^{H}\nabla_{\mathbf{x}_t}Q_{\bar{\phi}_k^h}(\mathbf{s},\mathbf{x}_t), \tag{18}$$

where $C = \mathbb{E}_{(\mathbf{s},\mathbf{a})\sim\mathcal{D}}|Q_{\bar{\phi}_k^h}(\mathbf{s},\mathbf{a})|$ is an estimated scaling constant that eliminates the influence of varying Q-value scales in different environments. We summarize the full algorithm of DAC for offline RL in Algorithm 1.

## 3.2 Policy extraction

We denote $\pi_\theta(\mathbf{a}|\mathbf{s})$ as the trained diffusion policy through denoisng process using noise predictor $\epsilon_\theta(\mathbf{x}_t,\mathbf{s},t)$. While $\pi_\theta(\mathbf{a}|\mathbf{s})$ is capable of generating the target policy, we aim to reduce the uncertainty of the denoising process during the evaluation phase. To achieve this, we sample a small batch of $N_a$ actions and select the action with the highest Q-ensemble mean value, resulting in better performance:

$$\pi(\mathbf{s}) = \arg\max_{\mathbf{a}_1,\ldots,\mathbf{a}_{N_a}\sim\pi_\theta(\cdot|\mathbf{s})}\mathbb{E}_h[Q_{\bar{\phi}_k^h}(\mathbf{s},\mathbf{a})]. \tag{19}$$

This approach is commonly employed in methods where a stochastic actor is trained for critic learning, and a deterministic policy is implemented during evaluation (Brandfonbrener et al., 2021; Haarnoja et al., 2018). Since $\pi_\theta(\mathbf{a}|\mathbf{s})$ is trained as a target policy, the sampling number $N_a$ can be relatively small. Through our experiments, we find that DAC can achieve superior performance with $N_a = 10$. In comparison, SfBC (Chen et al., 2022), Diffusion Q-learning (Wang et al., 2022) and IDQL (Hansen-Estruch et al., 2023) use $N_a = 32$, $N_a = 50$ and $N_a = 128$, respectively.

## 4 Related work

**Offline RL.** Recent research on offline RL often use value-based algorithms based on Q-learning or actor-critic learning (Sutton & Barto, 1999). Policy-regularization methods typically train a biased behavior sampler to help estimate the maximum Q-values within the behavior support. Among these approaches, BCQ (Fujimoto et al., 2019) learns a conditional-VAE (Sohn et al., 2015) to aid in sampling Q-learning targets; BEAR (Kumar et al., 2019) employs maximum mean discrepancy (MMD) to regularize the learned policy. Moreover, BRAC (Wu et al., 2019) is based on actor-critic learning framework and explores various regularization methods as value penalties; TD3+BC (Fujimoto & Gu, 2021) adds a behavior cloning term to regularize the learned policy in a supervised manner. Additionally, some methods implicitly regularize the policy by training pessimistic value-functions or using in-sample estimation. CQL (Kumar et al., 2020) learns conservative Q-values on OOD actions. IQL (Kostrikov et al., 2021) and IDQL (Hansen-Estruch et al., 2023) use asymmetric loss functions to approximate the maximal Q-value target via in-sample data. IVR (Xu et al., 2023) also employs in-sample learning while within the framework of behavior-regularized MDP problem, resulting in two implicit Q-learning objectives. Extreme Q-learning (Garg et al., 2023) estimates maximal Q-values using Gumbel regression. In addition to regularizing target policies to align with behavior policies, recent studies also emphasize the importance of regulating the steps of policy updates within the trust region (Schulman et al., 2015; 2017) to ensure policy improvements (Zhuang et al., 2023; Zhang & Tan, 2024; Lei et al., 2023). Our method conducts policy-regularization by focusing on KL-regularized policy iteration (Schulman et al., 2015; Peng et al., 2019; Nair et al., 2020), which regularizes the policy improvement step to fulfill the KL-divergence constraint, preventing the bootstrapping error of estimating Bellman targets.

**Diffusion models for offline RL.** Recent studies that utilize diffusion models for offline RL can be broadly categorized into two types: those that model entire trajectories and those that generate

Table 1: **Average normalized scores of DAC compared to other baselines.** We use the following abbreviations: "m" for "medium"; "r" for "replay"; "e" for "expert"; "u" for "umaze"; "div" for "diverse" and "l" for "large". For locomotion tasks, we use "v-2" version; for antmaze tasks, we use "v-0" version. We highlight in boldface the numbers within 5% of the maximal scores in each task. Furthermore, we also underline the highest scores achieved by prior methods.

| Dataset | Onestep-RL | CQL | IQL | IVR | EQL | Diffuser | DTQL | AlignIQL | SfBC | DQL | IDQL-A | DAC (ours) |
|---|---|---|---|---|---|---|---|---|---|---|---|---|
| halfcheetah-m | 48.4 | 44.0 | 47.4 | 48.3 | 48.3 | 44.2 | 57.9 | 46.0 | 45.9 | 51.1 | 51.0 | **59.1** ± 0.4 |
| hopper-m | 59.6 | 58.5 | 66.3 | 75.5 | 74.2 | 58.5 | **99.6** | 56.1 | 57.1 | 90.5 | 65.4 | **101.2** ± 2.0 |
| walker2d-m | 81.8 | 72.5 | 78.3 | 84.2 | 84.2 | 79.7 | 89.4 | 78.5 | 77.9 | 87.0 | 82.5 | **96.8** ± 3.6 |
| halfcheetah-m-r | 38.1 | 45.5 | 44.2 | 44.8 | 45.2 | 42.2 | 50.9 | 41.1 | 37.1 | 47.8 | 45.9 | **55.0** ± 0.2 |
| hopper-m-r | 97.5 | 95.0 | 94.7 | **99.7** | **100.7** | 96.8 | **100.0** | 74.8 | 86.2 | **101.3** | 92.1 | **103.1** ± 0.3 |
| walker2d-m-r | 49.5 | 77.2 | 73.9 | 81.2 | 82.2 | 61.2 | 88.5 | 76.5 | 65.1 | **95.5** | 85.1 | **96.8** ± 1.0 |
| halfcheetah-m-e | 93.4 | 91.6 | 86.7 | 94.0 | **94.2** | 79.8 | 92.7 | 89.1 | 92.6 | **96.8** | **95.9** | **99.1** ± 0.9 |
| hopper-m-e | 103.3 | 105.4 | 91.5 | **111.8** | **111.2** | 107.2 | 109.3 | 107.1 | 108.6 | **111.1** | 108.6 | **111.7** ± 1.0 |
| walker2d-m-e | **113.0** | 108.8 | 109.6 | 110.2 | **112.7** | 108.4 | 110.0 | 111.9 | 109.8 | 110.1 | **112.7** | **113.6** ± 3.5 |
| **locomotion total** | 684.6 | 698.5 | 749.7 | 749.7 | 752.9 | 678.0 | **798.3** | 681.1 | 680.3 | 791.2 | 739.2 | **836.4** |
| antmaze-u | 64.3 | 74.0 | 87.5 | 93.2 | 93.8 | - | **94.8** | **94.8** | 92.0 | 93.4 | 94.0 | **99.5** ± 0.9 |
| antmaze-u-div | 60.7 | **84.0** | 62.2 | 74.0 | **82.0** | - | 78.8 | 82.4 | **85.3** | 66.2 | 80.2 | **85.0** ± 7.9 |
| antmaze-m-play | 0.3 | 61.2 | 71.2 | 80.2 | 76.0 | - | 79.6 | 80.5 | 81.3 | 76.6 | **84.2** | **85.8** ± 5.5 |
| antmaze-m-div | 0.0 | 53.7 | 70.0 | 79.1 | 73.6 | - | 82.2 | **85.5** | 82.0 | 78.6 | 84.8 | **84.0** ± 6.2 |
| antmaze-l-play | 0.0 | 15.8 | 39.6 | 53.2 | 46.5 | - | 52.0 | **65.2** | 59.3 | 46.4 | 63.5 | 50.3 ± 8.6 |
| antmaze-l-div | 0.0 | 14.9 | 47.5 | 52.3 | 49.0 | - | **66.4** | 54.0 | 45.5 | 56.6 | **67.9** | 55.3 ± 10.3 |
| **antmaze total** | 125.3 | 303.6 | 378.0 | 432.0 | 420.9 | - | 441.4 | **474.8** | 445.4 | 417.8 | 474.6 | **459.9** |

behavior policies. Diffuser (Janner et al., 2022) trains a diffusion model as a trajectory planner. The policies are generated through guided-sampling with return prompts, similar to methods that modeling trajectories using Transformer (Chen et al., 2021; Janner et al., 2021). Diffusion Q-learning (Wang et al., 2022) employs a diffusion model as a biased behavior sampler, incorporating an additional loss to promote the denoised actions to achieve maximal Q-values. IDQL (Hansen-Estruch et al., 2023) and SfBC (Chen et al., 2022) use diffusion models to generate behavior policies. The target policies are extracted by re-sampling from diffusion-generated actions. Diffusion trusted Q-learning (Chen et al., 2024) also models behavior policies as diffusion models, while the diffusion-modeled behavior samplers are used to extract Gaussian policies that focus on high-reward modes. In comparison, DAC directly utilizes trained diffusion model as the target policy in the actor-critic paradigm, rather than using it as a behavior sampler to estimate Q-learning targets.

**Diffusion models for online RL.** Recent research also investigates diffusion policies for online RL. QSM (Psenka et al., 2023) learns diffusion policies by directly aligning with the Boltzmann distribution of Q-functions. DIPO (Yang et al., 2023) updates diffusion policies by adjusting action samples in the replay buffer. To support online exploration, (Wang et al., 2024) employs Gaussian mixture models to estimate densities and entropy. Although these methods also learn diffusion target policy, they lack policy-regularization mechanisms to tackle the unique challenges of offline RL.

## 5 EXPERIMENTS

In this section, we empirically demonstrate the effectiveness of our proposed method by comparing it with a comprehensive set of recent approaches on D4RL benchmarks (Fu et al., 2020). We also conduct ablations to show the efficacy of soft Q-guidance in achieving stable convergence and strong performance. Additionally, we include sensitivity analysis of the Q-ensemble size and action sample size to demonstrate its robustness. Due to space limitations, we place experimental details and ablation studies on LCB and other hyperparameters in Appendix B and C.

### 5.1 COMPARISONS ON OFFLINE RL BENCHMARKS

We compare our approach against an extensive collection of baselines that solve offline RL using various methods. For explicit policy-regularization methods, we consider one-step RL (Onestep-RL) (Brandfonbrener et al., 2021), which conducts a single step actor-critic learning. For value-constraint methods, we compare against Conservative Q-learning (CQL) (Kumar et al., 2020). For in-sample estimation of maximal value targets, we include Implicit Q-learning (IQL) (Kostrikov et al., 2021),

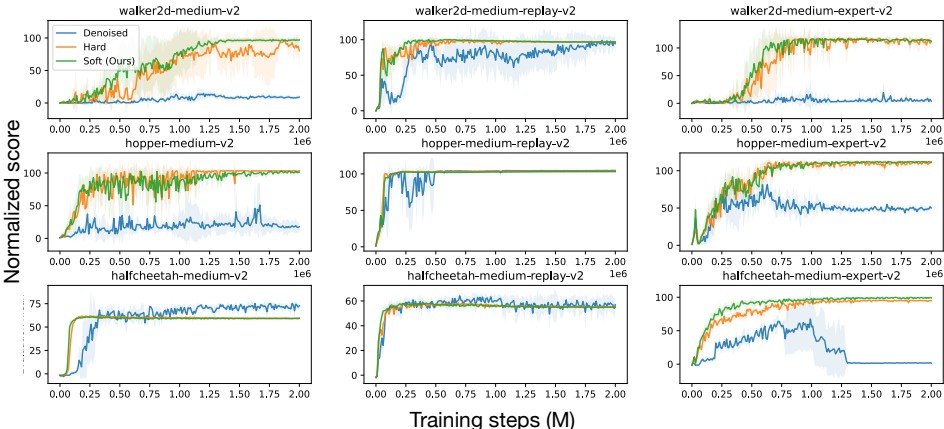

Figure 3: Training curves of DAC with different Q-gradient guidance. We compare soft Q-guidance (soft), hard Q-guidance (hard) and the denoised Q-guidance (denoised) on locomotion tasks. DAC with soft Q-guidance achieves stable convergence and strong performance across all the tasks.

Implicit Value Regularization (IVR) (Xu et al., 2023) and Extreme Q-learning (EQL) (Garg et al., 2023). Additionally, we compare our approach to methods that also learn diffusion policies. In this category, we compare against the most recent works including Diffuser (Janner et al., 2022), SfBC (Chen et al., 2022), Diffusion Q-learning (DQL) (Wang et al., 2022), Diffusion Trusted Q-Learning (DTQL) (Chen et al., 2024), AlignIQL (He et al., 2024) and IDQL (Hansen-Estruch et al., 2023). Specifically, we present the results of "IDQL-A" variant (Hansen-Estruch et al., 2023) for IDQL, which permits tuning of any amount of hyperparameters and exhibits strong performance. For the baselines, we report the best results from their own paper or tables in the recent papers (Wang et al., 2022; Chen et al., 2022; Hansen-Estruch et al., 2023). The main results are shown in Table 1. As for additional results on Adroit and Kitchen tasks, we refer readers to Appendix C.1.

Drawing from the experimental results, DAC outperforms prior methods by a significant margin across nearly all tasks. DAC significantly enhances the overall score on locomotion tasks, with an average increase of 5% compared to the best performance in prior studies. Notably, for "medium" tasks, where the dataset contains numerous sub-optimal trajectories, DAC consistently achieves improvements of over 10%. The antmaze domain presents a greater challenge due to the sparsity of rewards and the prevalence of sub-optimal trajectories. Consequently, algorithms must possess strong capabilities in stitching together sub-optimal subsequences to achieve high scores (Janner et al., 2022). It is evident that DAC outperforms or achieves competitive outcomes on antmaze tasks, with an almost perfect mean score ($\approx 100$) on the "antmaze-umaze" task. In the case of the most demanding "large" tasks, DAC performs comparably to previous methods, with the exception of IDQL-A, which consistently showcases superior performance. One potential explanation for this difference could be that we do not tune the rewards by subtracting a negative number, as is done in previous studies (Kumar et al., 2020; Kostrikov et al., 2021; Wang et al., 2022; Hansen-Estruch et al., 2023). This setting exacerbates the impact of reward sparsity in more intricate "large" environments, leading to slower convergence.

It is worth mentioning that we report the performance after convergence, which imposes a stronger requirement for evaluation, as it necessitates the model training to demonstrate the capability of convergence, rather than relying on online model selection (Wang et al., 2022; Kang et al., 2024). These requirements hold significant importance for ensuring robust deployment in real-world applications.

## 5.2 ABLATION STUDY ON Q-GRADIENT GUIDANCE

To demonstrate the effectiveness of soft Q-guidance, we compare DAC to two variants: one that utilizes hard Q-guidance by eliminating the noise scaling factor, and another that employs denoised Q-guidance by conducting guidance with denoised actions (see Appendix B for details). The training curves are presented in Figure 3, with the corresponding final normalized scores in Table 2. DAC with soft Q-guidance achieves the highest performance in nearly all tasks. The hard Q-guidance

Table 2: **Q-guidance ablation.** We compare soft Q-guidance against hard Q-guidance and denoised Q-guidance while maintaining the remaining settings the same, with the average normalized scores surpassing prior methods (Table 1) highlighted in boldface.

| Q-Target | walker2d | | | hopper | | | halfcheetah | | |
|---|---|---|---|---|---|---|---|---|---|
| | **m** | **m-r** | **m-e** | **m** | **m-r** | **m-e** | **m** | **m-r** | **m-e** |
| Denoised | $8.4 \pm 2.2$ | $95.4 \pm 7.5$ | $5.9 \pm 1.0$ | $17.8 \pm 8.2$ | $105 \pm 1.0$ | $49.5 \pm 1.4$ | **$71.9 \pm 1.9$** | **$56.7 \pm 5.3$** | $1.76 \pm 1.0$ |
| Hard | $85.2 \pm 16.1$ | **$96.9 \pm 0.5$** | $110.4 \pm 6.3$ | **$103.1 \pm 0.2$** | **$103.8 \pm 0.3$** | $110.2 \pm 2.4$ | **$59.5 \pm 0.5$** | **$55.3 \pm 0.4$** | $94.6 \pm 0.9$ |
| Soft (Ours) | **$96.8 \pm 3.6$** | **$96.8 \pm 1.0$** | **$113.6 \pm 3.5$** | **$101.2 \pm 2.0$** | **$113.1 \pm 0.3$** | **$111.7 \pm 1.0$** | **$59.1 \pm 0.4$** | **$55.0 \pm 0.2$** | **$99.1 \pm 0.9$** |

also performs well when the behavior dataset comprises an adequate number of optimal demonstrations. However, when confronted with tasks that involve numerous sub-optimal trajectories (such as "medium" datasets), the hard Q-guidance falls behind in comparison to the soft Q-guidance. Furthermore, the denoised Q-guidance often struggles to generate in-distribution actions and frequently fails. Nevertheless, it yields the highest score on "halfcheetah-m" task, which could be attributed to the fact that such task is more tolerant to OOD actions.

## 5.3 SENSITIVITY ANALYSIS

**Q-ensemble sizes.** We compare the performance of different Q-ensemble sizes in Figure 4. We find that ensemble sizes $\geq 5$ generally yield similar results, while using a number as few as two can reduce performance on certain tasks. This highlights the importance of accurately estimating the Q-gradient field in DAC for effective learning. Larger ensembles help smooth the NN-estimated Q-gradient fields, leading to more stable performance. We choose an ensemble size of 10 for all tasks to balance performance and computational cost.

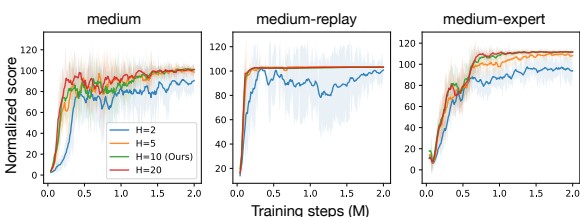

Figure 4: Experiments of using different Q-ensemble sizes $(H)$ on hopper tasks.

**Number of action samples.** Compared with methods that use diffusion models as behavior samplers, DAC directly models the target policy as a diffusion model, permitting fewer action samples to achieve good results. To demonstrate that DAC is robust to the choices of $N_a$, we conduct experiments with various $N_a$ on hopper tasks, as shown in Figure 5. We observe that increasing $N_a > 10$ gives similar results, while a small $N_a$ is detrimental to the performance by the randomness of denoising process of diffusion models.

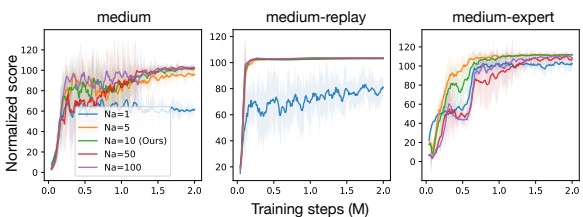

Figure 5: Experiments of using different number of action samples $(N_a)$ on hopper tasks.

## 6 CONCLUSION

In this paper, we propose the Diffusion Actor-Critic framework, which theoretically formulates the KL constraint policy iteration as a diffusion noise regression problem. The resulting policy improvement loss includes a soft Q-guidance term that adjusts the strength of Q-gradient guidance based on noise scales. This approach encourages the generation of high-reward actions while remaining within the behavior support. Furthermore, DAC avoids gradient propagation through the denoising path, significantly reducing training time for diffusion policies. Experiments demonstrate that our algorithm achieves stable convergence and superior performance on D4RL benchmarks, outperforming previous methods across nearly all tasks without the need for online model selection.

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

## A    PROOFS OF THEORETICAL RESULTS

**Lemma 1.** *Let $\epsilon \sim \mathcal{N}(\mathbf{0}, \mathbf{I})$ be a standard Gaussian noise, and $\mathbf{x}_t = \sqrt{\bar{\alpha}_t}\mathbf{a} + \sqrt{1 - \bar{\alpha}_t}\epsilon$ be the noise perturbation of the action $\mathbf{a}$ defined in (4). Then the denoising score function $\nabla_{\mathbf{x}_t} \log q_t(\mathbf{x}_t|\mathbf{a})$ maintains the property:*

$$\nabla_{\mathbf{x}_t} \log q_t(\mathbf{x}_t|\mathbf{a}) = -\frac{\epsilon}{\sqrt{1 - \bar{\alpha}_t}}, \tag{20}$$

*Proof.* Since $q_t(\mathbf{x}_t|\mathbf{x}_0) = \mathcal{N}(\mathbf{x}_t; \sqrt{\bar{\alpha}_t}\mathbf{x}_0, (1 - \bar{\alpha}_t)\mathbf{I})$, the density of noise distribution has the closed form:

$$q_t(\mathbf{x}_t|\mathbf{a}) \propto \exp\left[-\frac{(\mathbf{x}_t - \sqrt{\bar{\alpha}_t}\mathbf{a})^2}{2(1 - \bar{\alpha}_t)}\right]. \tag{21}$$

Therefore, we have

$$\begin{aligned}\nabla_{\mathbf{x}_t} \log q_t(\mathbf{x}_t|\mathbf{a}) &= -\frac{\mathbf{x}_t - \sqrt{\bar{\alpha}_t}\mathbf{a}}{1 - \bar{\alpha}_t} = -\frac{\sqrt{1 - \bar{\alpha}_t}\epsilon}{1 - \bar{\alpha}_t} \\ &= -\frac{\epsilon}{\sqrt{1 - \bar{\alpha}_t}}.\end{aligned} \tag{22}$$

$\square$

### A.1    PROOF OF THEOREM 1

*Proof.* Let $\epsilon \sim \mathcal{N}(\mathbf{0}, \mathbf{I})$ be a standard Gaussian noise. Consider the diffusion process $q_t(\mathbf{x}_t|\mathbf{a})$ defined in (4) using reparameterization trick. Then $\epsilon^*$ solves the noisy score matching objective:

$$\epsilon^* = \arg\min_{\tilde{\epsilon}} \mathbb{E}_{\mathbf{a} \sim \pi_{k+1}^*, t, \mathbf{x}_t \sim q_t(\cdot|\mathbf{a})}\left[\frac{1}{2}||\tilde{\epsilon}(\mathbf{x}_t, \mathbf{s}, t) + \sqrt{1 - \bar{\alpha}_t}\nabla_{\mathbf{x}_t} \log p_t^*(\mathbf{x}_t|\mathbf{s})||^2\right]. \tag{23}$$

We can rewrite the objective to obtain:

$$\epsilon^* = \arg\min_{\tilde{\epsilon}} \mathbb{E}_{\mathbf{a} \sim \pi_{k+1}^*, t, \mathbf{x}_t \sim q_t(\cdot|\mathbf{a})}, \left[\frac{1}{2}||\tilde{\epsilon}(\mathbf{x}_t, \mathbf{s}, t)||^2 + \tilde{\epsilon}(\mathbf{x}_t, \mathbf{s}, t) \cdot \sqrt{1 - \bar{\alpha}_t}\nabla_{\mathbf{x}_t} \log p_t^*(\mathbf{x}_t|\mathbf{s})\right] + C_1. \tag{24}$$

Consider the second term:

$$\begin{aligned}\mathbb{E}_{\mathbf{x}_t \sim p_t^*(\cdot|\mathbf{s})}&[\tilde{\epsilon}(\mathbf{x}_t, \mathbf{s}, t) \cdot \sqrt{1 - \bar{\alpha}_t}\nabla_{\mathbf{x}_t} \log p_t^*(\mathbf{x}_t|\mathbf{s})] \\ &= \sqrt{1 - \bar{\alpha}_t} \int_{\mathbf{x}_t} p_t^*(\mathbf{x}_t|\mathbf{s})\tilde{\epsilon}(\mathbf{x}_t, \mathbf{s}, t) \cdot \nabla_{\mathbf{x}_t} \log p_t^*(\mathbf{x}_t|\mathbf{s})d\mathbf{x}_t \\ &= \sqrt{1 - \bar{\alpha}_t} \int_{\mathbf{x}_t} \tilde{\epsilon}(\mathbf{x}_t, \mathbf{s}, t) \cdot \nabla_{\mathbf{x}_t} p_t^*(\mathbf{x}_t|\mathbf{s})d\mathbf{x}_t \\ &= \sqrt{1 - \bar{\alpha}_t} \int_{\mathbf{x}_t} \tilde{\epsilon}(\mathbf{x}_t, \mathbf{s}, t) \cdot \nabla_{\mathbf{x}_t} \int_{\mathbf{a}} q_t(\mathbf{x}_t|\mathbf{a})\pi^*(\mathbf{a}|\mathbf{s})d\mathbf{a}d\mathbf{x}_t \\ &= \sqrt{1 - \bar{\alpha}_t} \int_{\mathbf{x}_t} \tilde{\epsilon}(\mathbf{x}_t, \mathbf{s}, t) \cdot \int_{\mathbf{a}} q_t(\mathbf{x}_t|\mathbf{a})\nabla_{\mathbf{x}_t} \log q_t(\mathbf{x}_t|\mathbf{a})\pi^*(\mathbf{a}|\mathbf{s})d\mathbf{a}d\mathbf{x}_t \\ &= \int_{\mathbf{x}_t} \int_{\mathbf{a}} \tilde{\epsilon}(\mathbf{x}_t, \mathbf{s}, t) \cdot \sqrt{1 - \bar{\alpha}_t}\nabla_{\mathbf{x}_t} \log q_t(\mathbf{x}_t|\mathbf{a})q_t(\mathbf{x}_t|\mathbf{a})\pi^*(\mathbf{a}|\mathbf{s})d\mathbf{a}d\mathbf{x}_t \\ &= \mathbb{E}_{\mathbf{a} \sim \pi_{k+1}^*, \mathbf{x}_t \sim q_t(\cdot|\mathbf{a})}[\tilde{\epsilon}(\mathbf{x}_t, \mathbf{s}, t) \cdot \sqrt{1 - \bar{\alpha}_t}\nabla_{\mathbf{x}_t} \log q_t(\mathbf{x}_t|\mathbf{a})].\end{aligned}$$

Thus we obtain:

$$\begin{aligned}\epsilon^* &= \arg\min_{\tilde{\epsilon}} \mathbb{E}_{\mathbf{a} \sim \pi_{k+1}^*, t, \mathbf{x}_t \sim q_t(\cdot|\mathbf{a})}\left[\frac{1}{2}||\tilde{\epsilon}(\mathbf{x}_t, \mathbf{s}, t)||^2 + \tilde{\epsilon}(\mathbf{x}_t, \mathbf{s}, t) \cdot \sqrt{1 - \bar{\alpha}_t}\nabla_{\mathbf{x}_t} \log q_t(\mathbf{x}_t|\mathbf{a})\right] + C_1 \\ &= \arg\min_{\tilde{\epsilon}} \mathbb{E}_{\mathbf{a} \sim \pi_{k+1}^*, t, \mathbf{x}_t \sim q_t(\cdot|\mathbf{a})}\left[\frac{1}{2}||\tilde{\epsilon}(\mathbf{x}_t, \mathbf{s}, t) + \sqrt{1 - \bar{\alpha}_t}\nabla_{\mathbf{x}_t} \log q_t(\mathbf{x}_t|\mathbf{a})||^2\right] + C_1 - C_2 \\ &= \arg\min_{\tilde{\epsilon}} \mathbb{E}_{\mathbf{a} \sim \pi_{k+1}^*, t, \mathbf{x}_t \sim q_t(\cdot|\mathbf{a})}\left[\frac{1}{2}||\tilde{\epsilon}(\mathbf{x}_t, \mathbf{s}, t) - \epsilon||^2\right] + C_1 - C_2. \text{ (by Lemma 1)}\end{aligned}$$

Here $C_1$ and $C_2$ are constants independent of $\tilde{\epsilon}$, which completes the proof. $\square$

## A.2 PROOF OF THEOREM 2

*Proof.* Let the diffusion process $q_t(\mathbf{x}_t|\mathbf{a})$ be defined in (4) using reparameterization trick by sampling standard Gaussian noise $\boldsymbol{\epsilon} \sim \mathcal{N}(\mathbf{0}, \mathbf{I})$. We rewrite the training objective:

$$\arg\min_\theta \mathbb{E}_{(\mathbf{s},\mathbf{a})\sim\mathcal{D},t,\mathbf{x}_t\sim q_t(\mathbf{x}_t|\mathbf{a})}[\frac{1}{2}||\boldsymbol{\epsilon}_\theta(\mathbf{x}_t,\mathbf{s},t) - \boldsymbol{\epsilon}^*(\mathbf{x}_t,\mathbf{s},t)||^2]$$

$$= \arg\min_\theta \mathbb{E}_{(\mathbf{s},\mathbf{a})\sim\mathcal{D},t,\mathbf{x}_t\sim q_t(\mathbf{x}_t|\mathbf{a})}[\frac{1}{2}||\boldsymbol{\epsilon}_\theta(\mathbf{x}_t,\mathbf{s},t) + \sqrt{1-\bar{\alpha}_t}\nabla_{\mathbf{x}_t}\log p_t^*(\mathbf{x}_t|\mathbf{s})||^2]$$

$$= \arg\min_\theta \mathbb{E}_{(\mathbf{s},\mathbf{a})\sim\mathcal{D},t,\mathbf{x}_t\sim q_t(\mathbf{x}_t|\mathbf{a})}[\frac{1}{2}||\boldsymbol{\epsilon}_\theta(\mathbf{x}_t,\mathbf{s},t) + \sqrt{1-\bar{\alpha}_t}\nabla_{\mathbf{x}_t}\log p_t(\mathbf{x}_t|\mathbf{s})$$
$$+ \frac{1}{\eta}\nabla_{\mathbf{x}_t}Q^{\pi^k}(\mathbf{x}_t,\mathbf{s})||^2]$$

$$= \arg\min_\theta \mathbb{E}_{(\mathbf{s},\mathbf{a})\sim\mathcal{D},t,\mathbf{x}_t\sim q_t(\mathbf{x}_t|\mathbf{a})}[\frac{1}{2}||\boldsymbol{\epsilon}_\theta(\mathbf{x}_t,\mathbf{s},t)||^2 + \boldsymbol{\epsilon}_\theta(\mathbf{x}_t,\mathbf{s},t) \cdot \sqrt{1-\bar{\alpha}_t}\nabla_{\mathbf{x}_t}\log p_t(\mathbf{x}_t|\mathbf{s})$$
$$+ \boldsymbol{\epsilon}_\theta(\mathbf{x}_t,\mathbf{s},t) \cdot \frac{1}{\eta}\nabla_{\mathbf{x}_t}Q^{\pi^k}(\mathbf{x}_t,\mathbf{s})] + C_1$$

Similar to the proof of Theorem 1, we can rewrite the second term to obtain:

$$\mathbb{E}_{t,\mathbf{x}_t\sim p_t(\cdot|\mathbf{s})}[\boldsymbol{\epsilon}_\theta(\mathbf{x}_t,\mathbf{s},t) \cdot \sqrt{1-\bar{\alpha}_t}\nabla_{\mathbf{x}_t}\log p_t(\mathbf{x}_t|\mathbf{s})]$$

$$= \sqrt{1-\bar{\alpha}_t}\int_{\mathbf{x}_t} p_t(\mathbf{x}_t|\mathbf{s})\boldsymbol{\epsilon}_\theta(\mathbf{x}_t,\mathbf{s},t)(\mathbf{x}_t,\mathbf{s},t) \cdot \nabla_{\mathbf{x}_t}\log p_t(\mathbf{x}_t|\mathbf{s})d\mathbf{x}_t$$

$$= \sqrt{1-\bar{\alpha}_t}\int_{\mathbf{x}_t} \boldsymbol{\epsilon}_\theta(\mathbf{x}_t,\mathbf{s},t) \cdot \nabla_{\mathbf{x}_t}p_t(\mathbf{x}_t|\mathbf{s})d\mathbf{x}_t$$

$$= \sqrt{1-\bar{\alpha}_t}\int_{\mathbf{x}_t} \boldsymbol{\epsilon}_\theta(\mathbf{x}_t,\mathbf{s},t) \cdot \nabla_{\mathbf{x}_t}\int_{\mathbf{a}} q_t(\mathbf{x}_t|\mathbf{a})\pi_\beta(\mathbf{a}|\mathbf{s})d\mathbf{a}d\mathbf{x}_t$$

$$= \sqrt{1-\bar{\alpha}_t}\int_{\mathbf{x}_t} \boldsymbol{\epsilon}_\theta(\mathbf{x}_t,\mathbf{s},t) \cdot \int_{\mathbf{a}} q_t(\mathbf{x}_t|\mathbf{a})\nabla_{\mathbf{x}_t}\log q_t(\mathbf{x}_t|\mathbf{a})\pi_\beta(\mathbf{a}|\mathbf{s})d\mathbf{a}d\mathbf{x}_t$$

$$= \sqrt{1-\bar{\alpha}_t}\int_{\mathbf{x}_t}\int_{\mathbf{a}} \boldsymbol{\epsilon}_\theta(\mathbf{x}_t,\mathbf{s},t) \cdot \nabla_{\mathbf{x}_t}\log q_t(\mathbf{x}_t|\mathbf{a})q_t(\mathbf{x}_t|\mathbf{a})\pi_\beta(\mathbf{a}|\mathbf{s})d\mathbf{a}d\mathbf{x}_t$$

$$= \mathbb{E}_{\mathbf{a}\sim\pi_\beta,t,\mathbf{x}_t\sim q_t(\cdot|\mathbf{a})}[\boldsymbol{\epsilon}_\theta(\mathbf{x}_t,\mathbf{s},t) \cdot \sqrt{1-\bar{\alpha}_t}\nabla_{\mathbf{x}_t}\log q_t(\mathbf{x}_t|\mathbf{a})]$$

$$= \mathbb{E}_{\mathbf{a}\sim\pi_\beta,t,\mathbf{x}_t\sim q_t(\cdot|\mathbf{a})}[-\boldsymbol{\epsilon}_\theta(\mathbf{x}_t,\mathbf{s},t) \cdot \boldsymbol{\epsilon}] \text{ (by Lemma 1).}$$

Therefore, we have:

$$\arg\min_\theta \mathbb{E}_{(\mathbf{s},\mathbf{a})\sim\mathcal{D},t,\mathbf{x}_t\sim q_t(\mathbf{x}_t|\mathbf{a})}[\frac{1}{2}||\boldsymbol{\epsilon}_\theta(\mathbf{x}_t,\mathbf{s},t) - \boldsymbol{\epsilon}^*(\mathbf{x}_t,\mathbf{s},t)||^2]$$

$$= \arg\min_\theta \mathbb{E}_{(\mathbf{s},\mathbf{a})\sim\mathcal{D},t,\mathbf{x}_t\sim q_t(\mathbf{x}_t|\mathbf{a})}[\frac{1}{2}||\boldsymbol{\epsilon}_\theta(\mathbf{x}_t,\mathbf{s},t)||^2 + \boldsymbol{\epsilon}_\theta(\mathbf{x}_t,\mathbf{s},t) \cdot \sqrt{1-\bar{\alpha}_t}\nabla_{\mathbf{x}_t}\log q_t(\mathbf{x}_t|\mathbf{a})$$
$$+ \boldsymbol{\epsilon}_\theta(\mathbf{x}_t,\mathbf{s},t) \cdot \frac{1}{\eta}\nabla_{\mathbf{x}_t}Q^{\pi^k}(\mathbf{x}_t,\mathbf{s})] + C_1$$

$$= \arg\min_\theta \mathbb{E}_{(\mathbf{s},\mathbf{a})\sim\mathcal{D},t,\mathbf{x}_t\sim q_t(\mathbf{x}_t|\mathbf{a})}[\frac{1}{2}||\boldsymbol{\epsilon}_\theta(\mathbf{x}_t,\mathbf{s},t)||^2 - \boldsymbol{\epsilon}_\theta(\mathbf{x}_t,\mathbf{s},t) \cdot \boldsymbol{\epsilon}$$
$$+ \boldsymbol{\epsilon}_\theta(\mathbf{x}_t,\mathbf{s},t) \cdot \frac{1}{\eta}\nabla_{\mathbf{x}_t}Q^{\pi^k}(\mathbf{x}_t,\mathbf{s})] + C_1$$

$$= \arg\min_\theta \mathbb{E}_{(\mathbf{s},\mathbf{a})\sim\mathcal{D},t,\mathbf{x}_t\sim q_t(\mathbf{x}_t|\mathbf{a})}[\frac{1}{2}||\boldsymbol{\epsilon}_\theta(\mathbf{x}_t,\mathbf{s},t) - \boldsymbol{\epsilon} + \frac{1}{\eta}\nabla_{\mathbf{x}_t}Q^{\pi^k}(\mathbf{x}_t,\mathbf{s})||^2] + C_1 - C_2$$

$$= \arg\min_\theta \mathbb{E}_{(\mathbf{s},\mathbf{a})\sim\mathcal{D},t,\boldsymbol{\epsilon}\sim\mathcal{N}(\mathbf{0},\mathbf{I})}[\frac{1}{2}||\boldsymbol{\epsilon}_\theta(\mathbf{x}_t,\mathbf{s},t) - \boldsymbol{\epsilon} + \frac{1}{\eta}\nabla_{\mathbf{x}_t}Q^{\pi^k}(\mathbf{x}_t,\mathbf{s})||^2] + C_1 - C_2,$$

where $\mathbf{x}_t = \sqrt{\bar{\alpha}_t}\mathbf{a} + \sqrt{1-\bar{\alpha}_t}\boldsymbol{\epsilon}$ is given by the reparameterization trick. $C_1$ and $C_2$ are constants independent of $\boldsymbol{\epsilon}_\theta$, which completes the proof.

$\square$

## B    EXPERIMENTAL DETAILS

We train our algorithm for 2 million gradient steps in order to ensure model convergence. For each environment, we carry out 8 independent training processes, with each process evaluating performance using 10 different seeds at intervals of 10,000 gradient steps. This leads to a total of 80 rollouts for each evaluation. We report the average score of evaluations in the last 50,000 gradient steps without any early-stopping selection, which fairly reflects the true performance after convergence. We perform our experiments on two GeForce RTX 4090 GPUs, with each experiment taking approximately 4 hours to complete, including both the training and evaluation processes. Our code implementation is built upon the jaxrl (Kostrikov, 2021) code base.

### B.1    NETWORK ARCHITECTURE

We employ simple 3 layer MLP with hidden dimension of 256 and Mish (Misra, 2019) activation for both the actor and critic networks. To enhance training stability, we implement target networks for both actor and critic, which track the exponential moving average (EMA) of the training networks. Specifically, we initialize the target networks $\epsilon_{\bar{\theta}}$ and $Q_{\bar{\phi}_k^h}$ with the same seed as training networks $\epsilon_\theta$ and $Q_{\bar{\phi}_k^h}$ respectively. We update the target actor network $\epsilon_{\bar{\theta}}$ every 5 gradient steps while update the target critic networks $Q_{\bar{\phi}_k^h}$ after each gradient step to further ensure training stability.

### B.2    HYPERPARAMETERS

We maintain consistent hyperparameter settings for the diffusion models and networks across all tasks. The hyperparameter settings are as follows:

Table 3: Hyperparameters for all networks and tasks.

| Hyperparameter | Value |
|---|---|
| T (Diffusion Steps) | 5 |
| $\beta_t$ (Noise Schedule) | Variance Preserving (Song et al., 2020) |
| H (Ensemble Size) | 10 |
| B (Batch Size) | 256 |
| Learning Rates (for all networks) | 3e-4, 1e-3 (antmaze-large) |
| Learning Rate Decay | Cosine (Loshchilov & Hutter, 2016) |
| Optimizer | Adam (Kingma & Ba, 2014) |
| $\eta_{\text{init}}$ (Initial Behavior Cloning Strength) | [0.1, 1] |
| $\alpha_\eta$ (for Dual Gradient Ascent) | 0.001 |
| $\alpha_{\text{ema}}$ (EMA Learning Rate) | 5e-3 |
| $N_a$ (Number of sampled actions for evaluation) | 10 |
| $b$ (Behavior Cloning Threshold) | [0.05, 1] |
| $\rho$ (Pessimistic factor) | [0, 2] |

Regarding the pessimistic factor $\rho$, we empirically find that selecting the smallest possible value for $\rho$ without causing divergence in Q-value estimation yields good outcomes, as the learned target policy already avoids sampling OOD actions. This makes the tuning of $\rho$ to be relatively straightforward. In terms of policy-regularization, DAC controls the trade-off between behavior cloning and policy improvement using either a constant $\eta \equiv \eta_{\text{init}}$ or learnable $\eta$ by setting $b$ for dual gradient ascent (Algorithm 1). For locomotion tasks, we employ dual gradient ascent which dynamically adjust $\eta$ to fulfil the policy constraint. As for antmaze tasks, we choose constant $\eta \equiv \eta_{\text{init}}$ during the training process. Moreover, since different tasks involve varying action dimensions, we choose different hyperparameters for each task. We consider values of $\eta_{\text{init}} \in [0.1, 1]$, $b \in [0.05, 1]$ and $\rho \in [0, 2]$. We summarize the hyperparameter settings for the reported results in Table 4.

### B.3    VALUE TARGET ESTIMATION

For the critic learning, we need to sample $\mathbf{a}' \sim \pi_{\theta_k}(\cdot|\mathbf{s}')$ to estimate the value target in (17). To enhance the training stability, we samples $M = 10$ actions from $\pi_{\theta_k}(\cdot|\mathbf{s}')$ through denoising pro-

Table 4: Hyperparameters settings for tasks.

| Tasks | $b$ | $\eta$ | $\rho$ | Regularization Type |
|---|---|---|---|---|
| hopper-medium-v2 | 1 | - | 1.5 | Learnable |
| hopper-medium-replay-v2 | 1 | - | 1.5 | Learnable |
| hopper-medium-expert-v2 | 0.05 | - | 1.5 | Learnable |
| walker2d-medium-v2 | 1 | - | 1 | Learnable |
| walker2d-medium-replay-v2 | 1 | - | 1 | Learnable |
| walker2d-medium-expert-v2 | 1 | - | 1 | Learnable |
| halfcheetah-medium-v2 | 1 | - | 0 | Learnable |
| halfcheetah-medium-replay-v2 | 1 | - | 0 | Learnable |
| halfcheetah-medium-expert-v2 | 0.1 | - | 0 | Learnable |
| antmaze-umaze-v0 | - | 0.1 | 1 | Constant |
| antmaze-umaze-diverse-v0 | - | 0.1 | 1 | Constant |
| antmaze-medium-play-v0 | - | 0.1 | 1 | Constant |
| antmaze-medium-diverse-v0 | - | 0.1 | 1 | Constant |
| antmaze-large-play-v0 | - | 0.1 | 1.1 | Constant |
| antmaze-large-diverse-v0 | - | 0.1 | 1 | Constant |

cess. As for locomotion tasks, we calculate the average value $1/M \sum_i^M [Q_{\bar{\phi}_k^h}(\mathbf{s}', \mathbf{a}_i')]$ over the sampled actions to estimate the target Q-value. While for antmaze tasks, we use the maximum $\max_{\mathbf{a}_1', \ldots, \mathbf{a}_M'} [Q_{\bar{\phi}_k^h}(\mathbf{s}', \mathbf{a}_i')]$ to address the problem of reward sparsity, which is consistent with previous research (Wang et al., 2022).

### B.4 Q-GRADIENT GUIDANCE

To fairly assess the performance of different Q-guidance, as shown in the 2D-bandit example (Figure 2) and an ablation study in Section 5.2, we modify the actor learning loss of DAC while keeping all the remaining settings the same. In the case of soft Q-guidance, we use the original loss of actor learning for DAC (16). For the hard Q-guidance, we modify (16) to remove the noise scale factor:

$$\mathcal{L}_{\text{hard}}(\theta) = \mathbb{E}_{(\mathbf{s}, \mathbf{a}) \sim \mathcal{D}, \boldsymbol{\epsilon}, t} \left[ \eta || \boldsymbol{\epsilon}_\theta(\mathbf{x}_t, \mathbf{s}, t) - \boldsymbol{\epsilon} ||^2 + \boldsymbol{\epsilon}_\theta(\mathbf{x}_t, \mathbf{s}, t) \cdot \nabla_{\mathbf{x}_t} Q^{\pi_k}(\mathbf{s}, \mathbf{x}_t) \right]. \tag{25}$$

Regarding the denoised Q-guidance used in Diffusion Q-learning (Wang et al., 2022), we use the following denoised Q-guidance loss:

$$\mathcal{L}_{\text{denoised}}(\theta) = \mathbb{E}_{(\mathbf{s}, \mathbf{a}) \sim \mathcal{D}, \boldsymbol{\epsilon}, t} \left[ \eta || \boldsymbol{\epsilon}_\theta(\mathbf{x}_t, \mathbf{s}, t) - \boldsymbol{\epsilon} ||^2 + \mathbb{E}_{\mathbf{x}_0 \sim \pi_\theta} [Q^{\pi_k}(\mathbf{s}, \mathbf{x}_0)] \right], \tag{26}$$

where the denoised action $\mathbf{x}_0$ is obtained through the denoising process used in DDPM, and the gradient $\partial \mathcal{L}_{\text{hard}}(\theta) / \partial \theta$ will be back-propagated through the denoising path. All the Q-functions are re-scaled by an estimated constant $\frac{1}{\mathbb{E}_{(\mathbf{s}, \mathbf{a}) \sim \mathcal{D}} |Q^{\pi_k}(\mathbf{s}, \mathbf{a})|}$ to remove the influence of different Q-value scales.

### B.5 2-D BANDIT EXAMPLE

For the 2-D bandit example (Figure 2), we generate 400 sub-optimal behavior actions by drawing samples from patterns with Gaussian noises. The reward values for each action are determined by the distances between action points and $(0.4, -0.4)$, i.e., reward $\sim -\sqrt{(x - 0.4)^2 + (y + 0.4)^2} + \mathcal{N}(\mathbf{0}, 0.5\mathbf{I})$. Therefore, the majority of actions are sub-optimal and the estimated gradient field will tend to promote the generation of out-of-distribution (OOD) actions. However, given that the true reward values associated with OOD actions are agnostic to the learner, a well-performing policy learned through offline RL should not deviate significantly from the behavior support. This highlights the superiority of DAC in this protocol. To reproduce the results presented in Figure 2, we train 20,000 gradient steps with a batch size of 128, a learning rate of 1e-3, a diffusion step $T = 50$, and behavior cloning threshold $b = 1.3$ for all the methods.

### B.6 REWARD TUNING

We adhere to the reward tuning conventions for locomotion tasks in the previous research (Kostrikov et al., 2021), which is defined as:

$$\tilde{r} = 1000 \times \frac{r}{\text{maximal trajectory return} - \text{minimal trajectory return}}. \tag{27}$$

As for antmaze task, it faces the challenge of sparse rewards, with the agent receiving a reward of 1 upon reaching the goal and 0 otherwise. Previous methods typically subtracts a negative constants (such as -1) from the rewards to tackle the issue of reward sparsity (Kostrikov et al., 2021; Wang et al., 2022). However, we empirically find that DAC performs well for most tasks without the need for such reward tuning technique. In our experiments, we simply employ the same tuning method (27) as the one used for locomotion tasks, which in fact scales the rewards by 1,000 for antmaze environments. This tuning method does not effectively tackle the problem of sparse rewards, which could potentially result in the inferior performance of DAC on the "large" antmaze tasks.

## C ADDITIONAL EXPERIMENTAL RESULTS

### C.1 RESULTS ON ADROIT AND KITCHEN TASKS

We provide additional results of DAC on Kitchen and Adroit tasks from D4RL dataset. Here we also involve the reported results from Diffusion Q-learning (Diffusion QL) Wang et al. (2022) as comparison. The results are summarized in Table (5), and the training curves are show in Figure 10 and Figure 11. Given that Diffusion Q-learning reports the scores with online model selection, it might be unfair to make direct comparison. Nevertheless, we do observe that DAC outperforms Diffusion Q-learning in Adroit domain.

Table 5: Additional experiments on Adroit and Kitchen tasks. Similar to locomotion and antmaze tasks, we report the convergent performance of DAC by averaging the scores in the last 50,000 gradient steps. Given that Diffusion Q-learning reports the scores with online model selection, it might be unfair to make direct comparison. However, we still observe that DAC outperforms Diffusion Q-learning in Adroit domain.

| Tasks | CQL | IQL | Diffusion QL | DTQL | DAC |
|-------|-----|-----|--------------|------|-----|
| pen-human-v1 | 35.2 | 71.5 | 72.8 | 64.1 | **81.3** $\pm$ 4.9 |
| pen-cloned-v1 | 27.2 | 37.3 | 57.3 | **81.3** | 63.9 $\pm$ 7.3 |
| **Adroit Average** | 31.2 | 54.4 | 65.1 | **72.7** | **72.6** |
| kitchen-complete-v0 | 43.8 | 62.5 | 80.8 | **84.0** | 77.4 $\pm$ 4.7 |
| kitchen-partial-v0 | 49.8 | 46.3 | 60.5 | **74.4** | 50.0 $\pm$ 5.7 |
| kitchen-mixed-v0 | 51.0 | 51.0 | **62.6** | 60.5 | 60.2 $\pm$ 7.3 |
| **Kitchen Average** | 48.2 | 53.3 | 69.0 | **73.0** | 62.6 |

### C.2 PESSIMISTIC VALUE TARGETS.

To demonstrate the importance of the LCB target in balancing the overestimation and underestimation of value targets, we compare a variant of DAC on locomotion tasks, wherein the LCB target is replaced by the ensemble minimum (min) (Fujimoto et al., 2018; Fujimoto & Gu, 2021; Wang et al., 2022). The results are shown in Table 6 and Figure 8. It is noteworthy that the variants of DAC without the LCB target also achieve competitive performance compared to prior research in many tasks.

### C.3 ADDITIONAL SENSITIVITY ANALYSIS OF HYPERPARAMETERS

**Diffusion step** $T$. We conduct experiments of DAC with different diffusion steps on hopper environments, as shown in Table 7 and Figure 6. We find that increasing the diffusion steps gives similar performance. Furthermore, we observe that larger $T$ converges slower than smaller $T$, which contradicts the claim in Diffusion Q-learning. In our experiments, we use $T = 5$ for all the tasks, and despite the shorter diffusion path, it is sufficient to achieve strong results.

Table 6: **Ensemble Q-target ablation.** We compare LCB target against the minimum target (min). We also involve the best scores of the prior methods (SOTA) from Table 1 for comparison.

| Q-Target | walker2d | | | hopper | | | halfcheetah | | |
|---|---|---|---|---|---|---|---|---|---|
| | m | m-r | m-e | m | m-r | m-e | m | m-r | m-e |
| SOTA | 87.0 | 95.5 | 113.0 | 90.5 | 101.3 | 111.2 | 51.1 | 47.8 | 96.8 |
| Min | $83.9 \pm 0.22$ | $66.3 \pm 9.7$ | $110.0 \pm 0.2$ | $\mathbf{100.8} \pm 0.9$ | $102.9 \pm 0.4$ | $\mathbf{111.3} \pm 0.1$ | $49.3 \pm 0.3$ | $43.1 \pm 0.2$ | $43.2 \pm 0.1$ |
| **LCB (Ours)** | $\mathbf{96.8} \pm 3.6$ | $\mathbf{96.8} \pm 1.0$ | $\mathbf{113.6} \pm 3.5$ | $101.2 \pm 2.0$ | $\mathbf{103.1} \pm 0.3$ | $\mathbf{111.7} \pm 1.0$ | $\mathbf{59.1} \pm 0.4$ | $\mathbf{55.0} \pm 0.2$ | $\mathbf{99.1} \pm 0.9$ |

Table 7: Sensitivity analysis of different length of diffusion steps.

| Diffusion step $T$ | 5 (ours) | 10 | 20 |
|---|---|---|---|
| hopper-medium-v2 | 101.2 | 102.0 | 99.7 |
| hopper-medium-reply-v2 | 103.1 | 102.3 | 102.2 |
| hopper-medium-expert-v2 | 111.7 | 110.2 | 110.1 |

**Pessimistic factor** $\rho$. We compare the performance of DAC with different pessimistic factors $\rho$, which controls the pessimism of the LCB of value targets in (17). The results on walker2d tasks are shown in Figure 7 (right). A small $\rho$ usually leads to overestimation of value targets and cause the Q-networks to diverge to $+\infty$, resulting in failed learning (as seen with $\rho = 0.5$). While a larger $\rho$ may allow converge, underestimating value targets negatively impacts final performance. In our experiments, we empirically find that selecting the smallest $\rho$ that prevents divergence of the Q-functions is sufficient for strong results, making tuning $\rho$ straightforward in practice.

**Behavior cloning threshold** $b$. We also conduct a sensitivity analysis of $b$ which controls the strength of behavior cloning in walker2d environments, as shown in Figure 7 (left). Given the constraint $||\epsilon_\theta(\mathbf{x}_t, \mathbf{s}, t) - \boldsymbol{\epsilon}||^2 \leq b$, the agent is prone to generate behavior data with a smaller $b$.

## C.4 TRAINING TIME COMPARISON

DAC trains significantly faster than Diffusion Q-learning without requiring gradient back-propagation through the denoising process. This time-saving advantage is more pronounced as the number of diffusion steps $T$ increases. To verify this, we conducted a experiment to assess the average training time ratio of performing a single-step batch update in DAC compared to Diffusion Q-learning. The results are shown in Table 8. The experiment shows that when $T = 5$, DAC requires less than half the time needed by Diffusion Q-learning to perform one-step updates. While for a diffusion step as large as 100, Diffusion Q-learning is more than 18 times slower compared with DAC. Considering that Diffusion Q-learning is unlikely to match DAC's performance with just one-tenth or even one-third of the training steps, it suggests that DAC can be trained more quickly.

Table 8: Time ratio between soft Q-guidance and denoised Q-guidance for conducting one-step batch gradient descent with different lengths of diffusion steps.

| Diffusion step $T$ | 5 | 10 | 20 | 50 | 100 |
|---|---|---|---|---|---|
| Time ratio | 45.1% | 31.4% | 19.9% | 9.7% | 5.4% |

## C.5 ADDITIONAL TRAINING CURVES

We also involve additional training curves on D4RL tasks in Figure 9, Figure 10 and Figure 11.

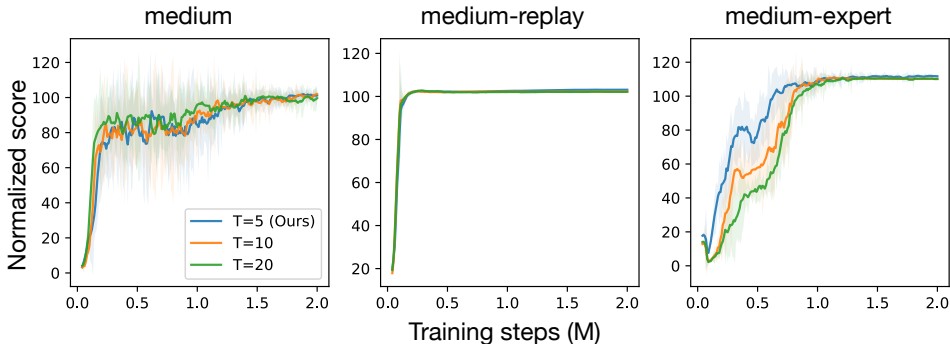

Figure 6: Sensitivity analysis of diffusion steps $T$ on hopper tasks.

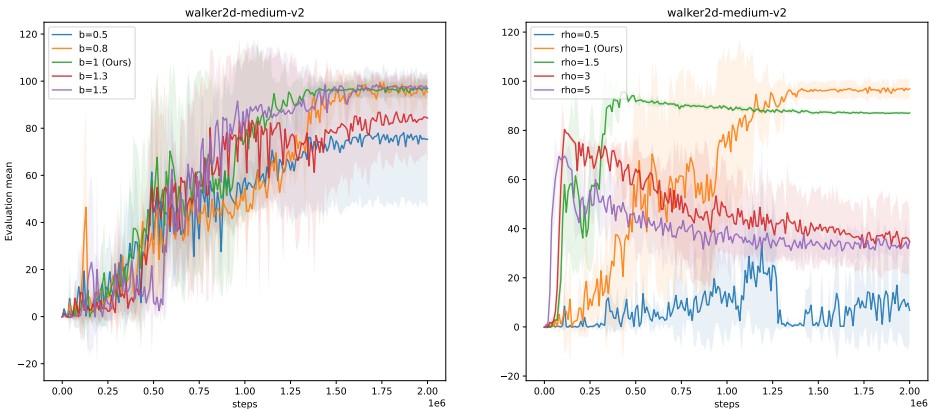

Figure 7: Sensitivity analysis of behavior cloning threshold $b$ (left) and pessimistic factor $\rho$ (right) on the walker2d-medium task.

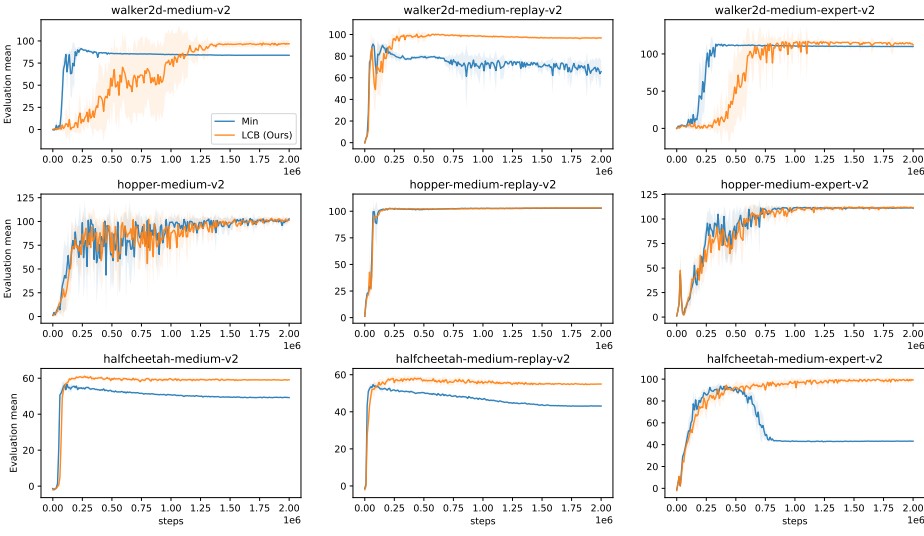

Figure 8: Value target ablation. We compare the LCB target against the target using ensemble minimum on locomotion tasks.

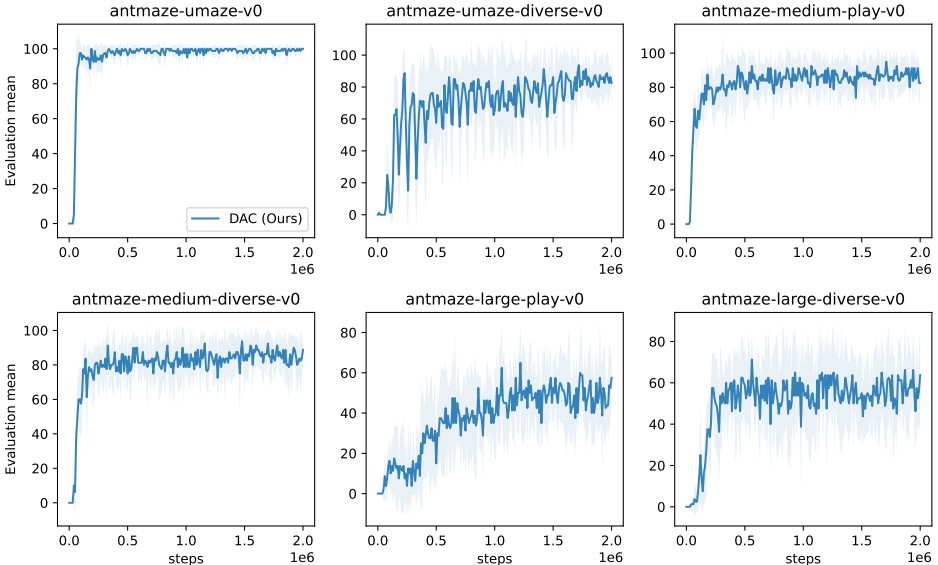

Figure 9: Training curves for antmaze tasks.

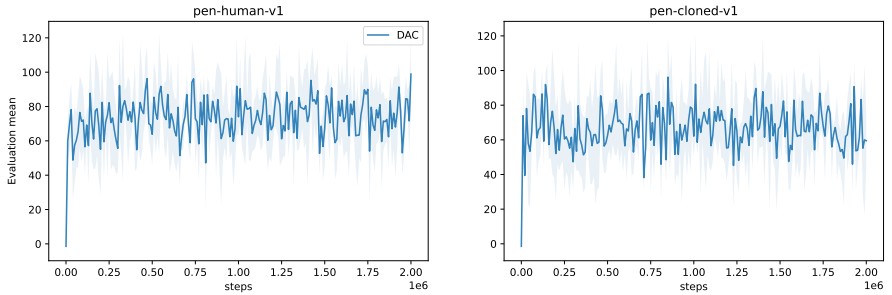

Figure 10: Training curves for Adroit tasks.

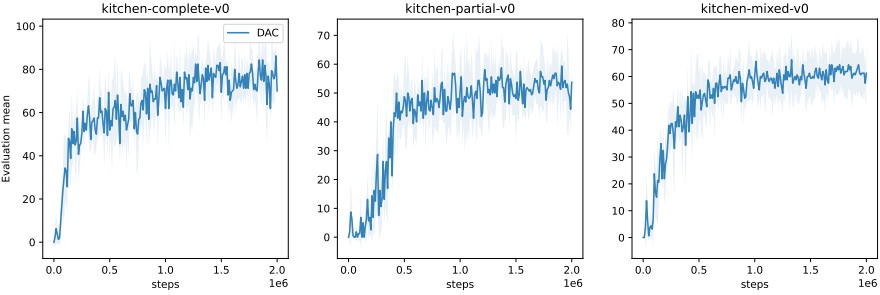

Figure 11: Training curves for Kitchen tasks.

