# OpenReview forum: "Diffusion Actor-Critic: Formulating Constrained Policy Iteration as Diffusion Noise Regression for Offline Reinforcement Learning"
_ICLR.cc/2025/Conference — ICLR 2025 Poster_

### Official Review · Reviewer_rDBF · 2024-10-24

**Soundness:** 3
**Presentation:** 3
**Contribution:** 2
**Rating:** 6
**Confidence:** 4

**Summary:**

The paper proposes DAC, an offline reinforcement learning algorithm that uses diffusion models to manage out-of-distribution actions and prevent the overestimation of value functions. DAC formulates the KL-constrained policy iteration as a diffusion noise regression problem, enabling the direct representation of target policies as diffusion models. The approach includes soft Q-guidance for actor training and a Q-ensemble for critic training, stabilizing Q-gradient estimation. DAC outperforms state-of-the-art methods on the D4RL benchmarks, demonstrating its effectiveness across various environments.

**Strengths:**

- The paper incorporates diffusion models into constrained policy iteration by using score functions, addressing the issue of the untraceable KL divergence in diffusion policies.

- The paper also presents extensive experiments and thorough ablation studies, demonstrating the effectiveness and robustness of DAC.

- DAC captures the multimodality of policies using diffusion model.

**Weaknesses:**

- There are some 2024 papers missing in the offline RL summaries (existing citations are 2023 at best), and accordingly I would like to see your results against them in the experimental section eg. AlignIQL: Policy Alignment in Implicit Q-Learning through Constrained Optimization (He et al.), Diffusion Policies creating a Trust Region for Offline Reinforcement Learning (Chen et al.).

- Recently, there is also a lot of work combining online RL and diffusion modelling, and I think a brief summary is also very necessary. Let me name a few famous works, you can add a paragraph to summarise eg. Policy Representation via Diffusion Probability Model for Reinforcement Learning(Yang et al.), Learning a Diffusion Model Policy from Rewards via Q-Score Matching(Psenka et al.), Diffusion Actor-Critic with Entropy Regulator(Wang et al.).

- I don't seem to have seen you perform ablation experiments for LCB, as well as ablation experiments for diffusion step size. Especially with the ablation experiments for LCB, I'd like to see the proportionality of the improvement in performance of LCB for your final performance compared to the improvement in performance of your proposed method.

- Experiments including $\rho$ were not analysed for sensitivity.

**Questions:**

- line244 or so, did you experiment with estimating entropy in the EDP?
- Why haven't you done experiments with Adroit Tasks and Kitchen Tasks and compared them with the latest algorithms? (It's not enough in the appendix, it's perfectly fine to put it in the body of the text for comparison)
- In Equation 15, the Q-function is well-trained only within the original data distribution. However, the distribution of $x_t$ tends to be more random and can significantly diverge from the dataset distribution. How can we ensure that the Q-function's gradients with respect to $x_t$ remain accurate and avoid substantial extrapolation errors?

---

> ### Author Response · Authors · 2024-11-21
> **Rebuttal by Authors (1/2)**
>
> >Q1. Missing some 2024 papers for comparison.
>
> A1. Thank you for highlighting those important papers. We will revise our experiment section to include comparisons with those latest methods. In conclusion, we find that our method outperforms AlignIQL and Diffusion Trusted Q-Learning in most environments, while AlignIQL performs better on the antmaze-large tasks, and Diffusion Trusted Q-Learning (DTQL) excels in Kitchen tasks. Here we summarize the scores as follows:
>
> Method|hop-m|hop-m-r|hop-m-e|wak-m|wak-m-r|wak-m-e|half-m|half-m-r|half-m-e
> |-|-|-|-|-|-|-|-|-|-
> |AlignIQL|56.1|74.8|107.1|78.5|76.5|111.9|46.0|41.1|89.1
> |DTQL|99.6|100.0|109.3|89.4|88.5|110.0|57.9|50.9|92.7
> |DAC|**101.2**|**103.1**|**111.7**|**96.8**|**96.8**|**113.6**|**59.1**|**55.0**|**99.1**
>
> Method|maze-u|maze-u-d|maze-m-p|maze-m-d|maze-l-p|maze-l-d
> |-|-|-|-|-|-|-
> |AlignIQL|94.8|82.4|80.5|**85.5**|**65.2**|**66.4**
> |DTQL|94.8|78.8|79.6|82.2|52.0|54.0
> |DAC|**99.5**|**85.0**|**85.8**|84.0|50.3|55.3
>
> Method|pen-h|pen-c|kitchen-c|kitchen-p|kitchen-m
> |-|-|-|-|-|-
> |AlignIQL|76.0|-|-|-|-
> |DTQL|64.1|**81.3**|**80.8**|**74.4**|60.2
> |DAC|**81.3**|63.9|77.4|50.0|**60.5**
>
> >Q2. A brief summary of papers combining online RL and diffusion modelling in a new paragraph is also very necessary.
>
> A2. Thank you for your valuable suggestions. We completely agree that those papers for online RL are also important for applying diffusion models for policy learning. We will add a separate paragraph to summarize those works to enhance the completeness of our literature review.
>
> >Q3. Don't seem to have seen you perform ablation experiments for LCB, as well as ablation experiments for diffusion step size. Experiments including $\rho$ were not analyzed for sensitivity.
>
> A3. We apologize for not clearly mentioning the additional experiments in the main text due to space constraints. We have included the ablation study of LCB as well as sensitivity analysis of hyperparameters, such as pessimistic factor $\rho$, diffusion steps $T$ and behavior cloning threshold $b$ in Appendix C. In our revision, we will revise the main text to help readers easily locate the additional experiments.
>
> For the ablation of the LCB target, we present the experimental results in Table 6, where the LCB target is replaced by the minimum target. The proportional improvement of LCB compared to the conventionally used minimum target is summarized as follows:
>
> |Tasks| min | LCB | Improvement
> |-|-|-|-
> |walker2d-m|83.9|96.8| + 15.4\%
> |walker2d-m-r|66.3|96.8| + 46.0\%
> |walker2d-m-e|110.0|113.6| + 3.3\%
> |hopper-m|100.8|101.2| + 0.4\%
> |hopper-m-r|102.9|103.1| + 0.2\%
> |hopper-m-e|111.3|111.7| + 0.4\%
> |halfcheetah-m|49.3|59.1| + 19.9\%
> |halfcheetah-m-r|43.1|55.0| + 27.6\%
> |halfcheetah-m-e|43.2|99.1| + 129.4\%
>
> We observe that the improvement from the minimum to LCB is quite significant, particularly in the walker2d and halfcheetah environments. These results support our initial motivation for selecting LCB rather than the conventionally used minimum as value targets: as the size of the Q-ensemble increases, taking the minimum across a larger number of Q-functions leads to overly pessimistic estimates. For instance, consider an ensemble of $n$ i.i.d. Q-functions $\{Q _ i\} _ {i=1}^n$. The probability that the ensemble minimum is less than a certain value $x$, i.e., $\mathbb{P}(\min \{Q _ i\} _ {i=1}^n< x)= 1 - (1- \mathbb{P}(Q _ i < x))^n$, increases exponentially with the ensemble size $n$, resulting in over-pessimism for larger ensembles.
>
> Regarding the pessimistic factor $\rho$, we have also involved the sensitivity analysis in Figure 7(right) in Appendix C. We use "walker2d-medium-v2" as an example, as we find that the training curves for other tasks are similar. Our experiments show that a small $\rho$ can lead to an overestimation of Q-functions, probably causing divergence in the estimated Q-values and resulting in failed learning. Conversely, a larger $\rho$ may underestimate the Q-values, leading to a lower final score. In our experiments, we empirically find that choosing the smallest $\rho \geq 0$ that does not cause divergence in the estimated Q-function is sufficient for achieving good results, making the tuning of $\rho$ quite straightforward.
>
> Additionally, we have also involved the sensitivity analysis of diffusion step size $T$ in Appendix C.3. We find that increasing the diffusion steps $T >5$ yields similar performance to $T = 5$. We choose $T = 5$ in our experiments to reduce the training and testing time, and this is sufficient to achieve good results.

---

> ### Author Response · Authors · 2024-11-21
> **Rebuttal by Authors (2/2)**
>
> >Q4. Why haven't you presented experiments with Adroit Tasks and Kitchen Tasks and compared them with the latest algorithms in the main text?
>
> A4. Thank you for your valuable feedback. We primarily aim to compare our method with the latest research that combines diffusion policies with offline reinforcement learning. To ensure a fair comparison without the need to reproduce results from scratch or conduct careful parameter tuning, we directly quote the reported scores from those papers. However, recent works on offline RL that use diffusion models and achieve strong results, such as Diffuser, SfBC, and IDQL in Table 1, do not include experiments on either Adroit tasks or Kitchen tasks. Therefore, we present the results on locomotion tasks and antmaze tasks, which are shared across all methods, to make the most of the space of the Table 1. To ensure the completeness of our experiments, we have also included the results for the Adroit and Kitchen tasks, although we have moved these results to the Appendix due to space constraints.
>
> >Q5. How can we ensure that the gradients of the Q-function with respect to $x _ t$ remain accurate and avoid significant extrapolation errors, especially since $x _ t$ tends to be more random and can diverge considerably from the dataset distribution?
>
> A5. Thank you for bringing this valuable problem into discussion. We answer your question from two aspects:
>
> * In fact, $Q(s, x _ t)$ is not necessary to be accurate in the whole extended action space (which is also impossible given the incompleteness of offline data coverage). Accurate $Q$ near the data is sufficient to generate high-quality rewarded actions. This is a key merit of our proposed soft Q-guidance in Section 3.1. As the soft Q-guidance reduces its influence to guide $x _ t$ towards the behavior support, $x _ t$ will be close to the data region in the final denoising steps, regardless of the accuracy of $Q(s, x _ t)$ in distant areas. And the accurately estimated $Q(s, x _ t)$ near the dataset is enough to generate high rewarded actions. As an evidence, you can check the estimated Q level curves (dash lines) in Fig.1, which are globally "wrong" for distant points, given the ground-truth should be concentric circles centered on (0.4, -0.4) (see configurations of 2D-bandit demo in Appendix B.5). Such inaccuracy of $Q(s, x _ t)$ on distant points does not affect the soft Q-guidance, but indeed cause other guidances to produce OOD actions.
>
> * Legal actions are typically compact. For instance, in D4RL, each action dimension is limited to [-1, 1], and actions outside this range are clipped before being sent to the environment. Such clipping of action is also conventionally applied to $x _ t$ during the denoising process (see Diffusion QL or IDQL for examples). This clipping prevents $x _ t$ to be far from the behavior set, alleviating the issue of wrongly estimated $Q(s, x _ t)$ on distant $x _ t$. However, our experiments find that action clipping is not the deterministic factor addressing such problem compared to the soft Q-guidance, as demonstrated in Figure 2 where we report results without action clipping.
>
> >Q6. In line 244 or so, did you experiment with estimating entropy in the EDP?
>
> A6. Since our diffusion policy cannot directly generate explicit density functions, we did not estimate the entropy of the policies in our examples. Instead, we simply visualized the sampled policy actions in Figure 2. Regarding the "estimating entropy in the EDP," we apologize for not fully understanding your question. Could you please clarify it further, especially the meaning of "EDP"? We would be happy to provide a detailed response.

---

> > ### Comment · Reviewer_rDBF · 2024-11-21
> > **Response to rebuttal**
> >
> > Thanks for your reply, which solved part of my problem. I observed that the performance gap between DAC and DTQL is large (some good, some bad) in pen-h, pen-c, kitchen-c, and kitchen-p. I want to see how you explain this phenomenon.
> >
> > Try not to use the expression "what will you do" in the rebuttal process. For ICLR, you need to modify the pdf directly and indicate where you modified it.
> >
> > Regarding the experiment of diffusion step length, did you conduct experimental results with less than 5 steps? Is it possible to achieve good performance with 3 steps?
> >
> > "EDP" is https://arxiv.org/abs/2305.20081. What effect can you get in your work by using the simplest method to estimate entropy? (Try it in a few of the most representative environments)

---

> > > ### Author Response · Authors · 2024-11-23
> > > **Thank you for your reply**
> > >
> > > Dear Reviewer rDBF:
> > >
> > > Thank you for your reply. We have revised our PDF to include the important papers and baselines you mentioned. Specifically, we summarize the revisions as follows:
> > >
> > > * We add AlignIQL and DTQL for comparison in the experiment section (Table 1), along with a brief summary of DTQL on line 406. We also included DTQL results in Table 5 for the Adroit and Kitchen tasks.
> > > * We add a new paragraph reviewing papers that combine diffusion models with online RL on line 410-416. Since our focus is on offline RL, we include only the mentioned papers and provide a concise introduction.
> > > * We also add a brief introduction of three papers mentioned by Reviewer GEf6 on line 370-373 to enhance the completeness of our review.
> > > * We have shortened the descriptions of Figure 1 for conciseness and to allow space for new content. Additionally, we revise lines 422-424 to help readers easily find the additional experiments, including ablations on LCB.
> > >
> > > >Q7. Explain why the performances of DAC and DTQL differ significantly on the Adroit and Kitchen tasks.
> > >
> > > Thank you for bringing this interesting phenomenon into the discussion. While we can't provide a definitive answer without thorough experiments, we can offer a possible explanation based on the mechanisms used by DAC and DTQL. Unlike locomotion tasks, the Adroit and Kitchen tasks have higher action dimensions and sparser reward feedback. The trajectories themselves display multimodal characteristics (as shown in Figures 2 and 3 in [1]). For example, the Kitchen dataset includes trajectories for four subtasks: opening the microwave, moving the kettle, flipping the light switch, and sliding open the cabinet door. This multimodality, combined with sparse rewards, may create distant and compact high-reward modes in the action space, with low rewards in between. DTQL uses Gaussian policies to randomly target one high-reward mode at a time, avoiding the risk of generating artifacts that connect these modes. In contrast, DAC directly trains diffusion policies, attempting to recover all high-reward modes, which can lead to hallucinations of high-reward actions between modes, especially when the diffusion step (T) is relatively low (see Figure 2 of DiffusionQL or Figure 2 of our paper for illustrations). Such insufficient learning of multi-modality could possibly be alleviated by using a larger diffusion step $T$. For instance, we observe that $T=5$ is difficult to tune for the Kitchen-c task, so we applied $T=50$ for Kitchen-c while keeping $T=5$ for other tasks.
> > >
> > > >Q8. Is it possible to achieve good performance with 3 steps?
> > >
> > > We conducted additional experiments with a diffusion step $T=3$ on locomotion tasks for illustration. We find that $T=3$ achieves good results in most tasks, only slightly worse than $T=5$. While $T=3$ performs worse than $T=5$, it remains competitive with other baselines.
> > >
> > > | Diffusion step $T$ | 3 | 5(ours)
> > > | - | - | -
> > > | walker2d-m | 91.3 | 101.2
> > > | walker2d-m-r | 93.1 | 103.1
> > > | walker2d-m-e | 113.3  | 111.7
> > > | hopper-m | 99.9 | 101.2
> > > | hopper-m-r | 103.0 | 103.1
> > > | hopper-m-e| 100.3  | 111.7
> > > | halfcheetah-m | 57.3 | 59.1
> > > | halfcheetah-m-r | 53.8 | 55.0
> > > | halfcheetah-m-e | 93.0 | 99.1
> > >
> > > >Q9. Experiment with estimating entropy in the EDP.
> > >
> > > We sincerely appreciate your suggestion to explore entropies for a deeper understanding of various diffusion policy methods. For entropy estimation, we could use Gaussian mixture model estimation with the EM algorithm, as seen in *Diffusion Actor-Critic with Entropy Regulator (Wang et al.)*. Since we didn't compare EDP with our methods in the paper, it's challenging to reproduce the EDP results and entropy estimation due to time limitations. Our primary focus is to highlight the performance of our methods using convergent scores and training curves. We will explore other aspects of the learned policies, such as entropy, in our future research.
> > >
> > > [1] Venkatraman, Siddarth, et al. "Reasoning with latent diffusion in offline reinforcement learning." arXiv preprint arXiv:2309.06599 (2023).

---

> > > > ### Comment · Reviewer_rDBF · 2024-11-24
> > > >
> > > > Thanks for the reply, I'll keep it at a 6 but raise the confidence

---

> ### Author Response · Authors · 2024-11-24
> **Thank you for your feedback**
>
> We sincerely appreciate your detailed comments and suggestions. We also thank you for raising insightful questions during the discussion, which have greatly helped us deepen our understanding of recent research. Thank you once again!

---

### Official Review · Reviewer_sKYu · 2024-10-30

**Soundness:** 3
**Presentation:** 2
**Contribution:** 2
**Rating:** 6
**Confidence:** 3

**Summary:**

This paper presents Diffusion Actor-Critic (DAC), a novel offline reinforcement learning algorithm designed to directly generate target policies using diffusion models. DAC formalizes the policy iteration problem with Kullback-Leibler (KL) constraints as a diffusion noise regression problem, achieving policy constraints through soft Q guidance, thus avoiding the generation of actions outside the offline data distribution. The paper demonstrates the outstanding performance of DAC in D4RL benchmark tests, surpassing current state-of-the-art methods in nearly all environments, while showing stable convergence and robust performance.

**Strengths:**

Innovativeness: Introduces a new algorithm DAC and proposes the soft Q guidance, addressing several challenges in offline reinforcement learning.

Experimental Performance: Demonstrates outstanding performance in D4RL benchmark tests, especially surpassing current state-of-the-art algorithms in locomotion and antmaze tasks.

Theoretical Support: Provides clear theoretical justification by rigorously transforming the KL constrained policy iteration problem into a diffusion noise regression problem.

**Weaknesses:**

Lack of presentation: structure of the article is chaotic, failing to highlight key points, which creates some difficulty in reading.

Scope of Application: Although DAC performs well on standard benchmarks, its generalization ability in large-scale or complex real-world tasks has not been fully validated.

The advantages of Soft Q-guidance are not sufficiently pronounced compared to other methods.

**Questions:**

How does DAC's generalization performance hold up in larger-scale real-world applications, such as autonomous driving or industrial robotics? Is there a plan to further test the scalability of this method?

Does the trade-off in soft Q guidance choice show differences in performance across various environments? Can the specific effects of different weight parameters on model performance be further explored?

---

> ### Author Response · Authors · 2024-11-21
> **Rebuttal by Authors (1/2)**
>
> >Q1. For the weakness that the key points are hard to follow.
>
> A1. We apologize for any confusion in our presentation. In offline RL, we aim to learn policies using the available offline dataset without interacting with the environments, similar to situations like autonomous driving where online learning is costly or even impossible. However, learning policies beyond the behavior data requires querying the value function for actions not present in the dataset. This evaluation of out-of-distribution (OOD) actions increases the bootstrapping error in value function estimation, posing a major challenge in offline RL. To tackle this issue, policy-regularized algorithms keep the learned policy close to the behavior policy, and our proposed method falls into this category.
>
> While recent advances show that diffusion models effectively model behavior policies to enhance performance, directly modeling target policies as diffusion models is still under-explored in offline RL. Additionally, since density functions cannot be directly inferred through diffusion models, applying policy-regularized methods to diffusion policies is not straightforward. Our main contribution is demonstrating that a closed-form solution exists for modeling diffusion policies under the KL-constraint (Section 3), as theoretically supported by Theorems 1 and 2. Experiments reveal that our proposed method achieves high scores on commonly used benchmarks and exhibits stable convergence. We thank you for your valuable feedback and will polish our presentation structure to make it easier for readers to follow.
>
> >Q2. How does DAC's generalization performance perform in larger-scale real-world applications? Are there plans to test its scalability further?
>
> A2. Thank you for your valuable suggestion! We completely agree that validating efficacy in more complex real-world applications is essential for broader impact. However, in this paper, we focus solely on prototyping a general diffusion policy for continuous control, following the conventions of offline RL research. Testing in areas like autonomous driving or industrial robotics will be reserved for future research.
>
> >Q3. What are the specific effects of different weight parameters on model performance, particularly regarding the strength of soft Q-guidance across various environments?
>
> A3. We have conducted a sensitivity analysis on the intensity of the soft Q-guidance in Appendix C.3, as illustrated in Figure 7 (left). We use "walker2d-medium-v2" as an example since other environments exhibit similar patterns. In Figure 7 (left), we use the behavior cloning threshold $b$ to trade-off between policy cloning and policy improvement. Given that the constraint $\|\epsilon _ \theta(x _ t, s, t) - \epsilon \| \leq b$ is applied to regularize the learned diffusion policy (line 11, Algorithm 1), a smaller $b$ implies a stronger constraint, and the target policy becomes closer to the behavior policy (such $b=0.5$), resulting in reduced performance. On the other hand, the performance may deteriorate with a too large $b$ as the training process may become instable (such as $b=1.3$). Additionally, we have also included sensitivity analysis of other parameters to validate the efficacy of our method, such as Q-ensemble size (Section 5.3), diffusion step $T$ (Appendix C.3) and pessimistic factor $\rho$ for LCB (Appendix C.3 and Figure 7).

---

> ### Author Response · Authors · 2024-11-21
> **Rebuttal by Authors (2/2)**
>
> >Q4. For the weakness: the advantages of soft Q-guidance are not sufficiently pronounced compared to other methods.
>
> A4. We would like to demonstrate the advantages of soft Q-guidance both theoretically and experimentally.
>
> * The proposed soft Q-guidance is the first theoretically sound method in offline RL to directly learn diffusion models to represent the target policy rather than modeling the behavior policy. This approach has two key advantages:  first, diffusion policies can capture the complexities of the target polices, where multi-modality is also common (as illustrated in the second plot of Figure 2). Second, this representation serves as a closed-form solution to the KL-constraint policy iteration, theoretically addressing the optimization problem without requiring any density estimation, which enhances our understanding of diffusion policies. Furthermore, by framing the learning process as a score matching problem, we eliminate the need to back-propagate Q-gradients through the entire denoising process in Diffusion QL. This significantly reduces training time and allows for modeling with much larger diffusion steps $T$ (see Table 8 for experiments on time savings).
>
> * In terms of experimental performance, our method demonstrates strong results compared to recent baselines that also use diffusion models for offline RL. For example, DAC significantly enhances the overall score on locomotion tasks, with an average increase of over 5% compared to DiffusionQL. Given that baseline performances are already saturated in this area, this represents a clear improvement. Additionally, soft Q-guidance not only delivers strong results but also shows stable convergence, as confirmed by the training curves in Figure 3. It’s important to note that we report the average scores from evaluations during the last 50,000 gradient steps without any *online-model-selection* (OMS), which sets a higher protocol than DiffusionQL, which uses OMS to track the best models throughout the training process. If we were to apply OMS to our method, it could achieve even higher scores (please refer to the table in our response to Reviewer GEf6's Q3 for an example).

---

> > ### Comment · Reviewer_sKYu · 2024-11-21
> > **response to authors**
> >
> > The author's rebuttal effectively addresses my concerns, and I will raise my score to 6 (slightly above the acceptance line). Reason for not having a higher score: No more complex and challenging environment for the experiment

---

> > > ### Author Response · Authors · 2024-11-21
> > > **Thank you for your feedback**
> > >
> > > We appreciate your thorough review and suggestions. We will focus our future efforts on addressing more challenging real-world tasks to enhance our contributions to practical applications. Thank you once again!

---

### Official Review · Reviewer_Wqqq · 2024-11-03

**Soundness:** 3
**Presentation:** 3
**Contribution:** 3
**Rating:** 8
**Confidence:** 3

**Summary:**

This paper presents the Diffusion Actor-Critic (DAC) framework, aimed at addressing key challenges in offline reinforcement learning, particularly in policy optimization and high-reward action generation. By innovatively transforming the KL-constrained policy iteration problem into a diffusion noise regression problem, DAC introduces a soft Q-guidance mechanism. Through comparative experiments, DAC demonstrates superior performance over existing methods on multiple D4RL benchmark tasks, especially when handling datasets containing a large number of suboptimal trajectories. The experimental results indicate that DAC not only exhibits stable convergence but also surpasses other methods in overall scores, showcasing its potential in the field of offline reinforcement learning.

**Strengths:**

Originality:

The DAC framework which combines diffusion models with policy optimization in reinforcement learning is novel. By introducing soft Q-guidance, the paper effectively overcomes the problem of overestimating Q-values in offline learning, enhancing the stability of policy improvement. This approach offers a new direction for future research.

Significance:

The authors conduct extensive experiments on D4RL benchmarks, covering a variety of tasks with results that are clear and convincing. The experimental results is strong

Clarity:

The paper is clearly written with illustrative examples

**Weaknesses:**

The authors should consider evaluate DAC in more offline reinforcement learning tasks, particularly in real-world application areas such as robotic control and atari games. And It would be better to provide specific choice of hyperparameter configuration

**Questions:**

- Soft Q-guidance relies on the accurate estimation of Q-values. If Q-values are overly optimistic or pessimistic, how does this affect the generated final policy? Have you considered the propagation of such uncertainty in the learning process? If the weight adjustment for soft Q-guidance is improper, could it lead to an over-reliance on Q-values in the generated policy, potentially resulting in overfitting?

- Soft Q-guidance performs well in handling tasks, but that hard Q-guidance significantly deteriorates in performance when facing a large number of sub-optimal trajectories. Have you considered combining different guidance strategies to enhance the robustness of the algorithm? Specifically, have you evaluated the introduction of additional regularization methods in specific tasks to prevent the policy from overly relying on sub-optimal trajectories, thereby enhancing the model's stability and generalization ability?

- In the antmaze task, the author noted that sparse rewards have a certain impact on the algorithm's performance. In this regard, have you designed more effective reward mechanisms or exploration strategies to address the challenges posed by sparse rewards? In practical applications, if the algorithm encounters sparse rewards, what measures do you anticipate taking to adjust the DAC algorithm to ensure it can still effectively learn and optimize the policy?

- The authors mentioned using a lower confidence bound to counteract overly pessimistic estimates. Could this lead to a systematic bias in the Q-values? How significant is this bias in different task settings?

- How does the $\eta$ parameter in the objective function influence the optimization process? In certain scenarios, the choice of $\eta$ might lead to oscillations or divergence of the algorithm. Is there a need for a sensitivity analysis regarding this choice? If gradient vanishing issues arise, particularly when the update frequency of the target policy is high, how should this potential problem be addressed?

---

> ### Author Response · Authors · 2024-11-21
> **Rebuttal by Authors (1/3)**
>
> >Q1. For the weakness
>
> A1. Thank you for your valuable suggestions! In this paper, we focus solely on prototyping a general diffusion policy for continuous control, following the conventions of offline RL research. We completely agree that validating efficacy in more complex real-world applications is essential for broader impact. We will validate our method on more complex tasks in the future work.
>
> Additionally, as for the training results in the experiment section, we have involved the configuration of hyper-parameters in Appendix B. We have also included sensitivity analysis of hyper-parameters and the discussions in Appendix C, which may help tune the model for any new tasks.
>
> >Q2. How do overly optimistic or pessimistic Q-function estimates affect the performance of the final policy? Have you taken into account the propagation of this uncertainty during the learning process? Could improper weight adjustments for soft Q-guidance lead to an over-reliance on Q-values?
>
> A2. Thank you for bringing us those enlightening questions! We answer them one by one:
>
> * With enough training steps, overly optimistic or pessimistic Q-function estimates often occur for out-of-distribution (OOD) actions, as the value estimation for in-sample actions can usually be corrected by the available data. When the Q-function is overly optimistic for OOD actions, the final policy tends to generate actions that lie on the boundary of the behavior support, focusing on the edges where the Q-function believes it is better to venture out (similar to the demo in Figure 2). In contrast, an overly pessimistic Q-function for OOD actions may lead the final policy to seek a better policy strictly within the behavior support, as the overly pessimistic Q-function acts like another behavior policy regularizer that tends to produce in-sample actions.
>
> * Yes, we have considered the the propagation of this uncertainty during the training process. Since we learn the Q-functions through Bellman updates, inaccurately estimated value targets can be propagated through bootstrap sampling. To address this issue, we track the exponential moving average (EMA) of the running Q-functions with a learning rate of 0.005, which changes much slowly and stably. Using the EMA of Q-functions for value target estimation, further smoothed by the Q-ensemble, stabilizes the learning process and leads to convergence of Q-functions.
>
> * Yes, improper weight adjustments for soft Q-guidance may cause the generated policies overfit the Q-functions. Imagine an extreme cause that $\eta=0$, meaning the behavior cloning term is completely discarded. In this scenario , the target policy essentially replicates the score field of the Q-functions, and the action generation process becomes a gradient ascent along the Q-surface. As a result, the actions would cluster around the optima of the Q-surface, regardless of the behavior support. In our method, $\eta$ acts as a trade-off between behavior cloning and policy improvement, in line with conventions in offline reinforcement learning research.

---

> ### Author Response · Authors · 2024-11-21
> **Rebuttal by Authors (2/3)**
>
> >Q3. Have you considered combining different guidance strategies to enhance the robustness of the algorithm? Specifically, have you evaluated the introduction of additional regularization methods in specific tasks to prevent the policy from overly relying on sub-optimal trajectories ?
>
> A3. Thank you for your valuable suggestions! Here are our responses:
>
> * We primarily focus on the soft guidance as it is theoretically derived from the KL-constraint policy iteration. Additionally, when $t$ is close to $T$, the soft weight $\sqrt{1-\bar{\alpha} _ t} \approx 1$, causing soft guidance to closely resemble hard guidance in the early stages of denoising generation. As a result, we did not combine different guidance strategies in our paper and allowed the guidance strength to be automatically regulated by the soft weight. However, we acknowledge that other forms of guidance can also empirically perform well with function approximations, as illustrated in some tasks in Figure 3. It is possible that incorporating different types of guidance throughout the denoising process could stabilize the algorithm. We will explore this in our future work.
>
> * The main mechanism that we apply to prevent the policy from overly relying on sub-optimal trajectories is to adjust the weight of behavior cloning term, namely $\eta$ in the actor learning loss. We also offer an alternative regularization method that learns $\eta$ through dual gradient ascent, ensuring that $\|\epsilon _ \theta (x _ t , s, t) − \epsilon\|^2 \leq b$ for some threshold $b>0$. Both a smaller $\eta$ and a larger $b$ work well to improve significantly over behavior trajectories. Additionally, we could apply further regularization to value functions, such as the methods used in CQL or IQL. In our early research, we empirically found that using CQL regularization on value functions does not enhance performance, possibly due to the lack of smoothness in CQL-learned value functions for out-of-distribution generation, which leads to unstable Q-gradients for diffusion guidance. We plan to explore other regularization methods in our future research.
>
> >Q4. Have you designed more effective reward mechanisms or exploration strategies to tackle the challenges of sparse rewards? What measures do you anticipate taking to modify the DAC algorithm to ensure it can still effectively learn in sparse reward tasks?
>
> A4. Essentially, the impact of sparse rewards on our algorithm's performance stems from the inherent challenges of learning smooth and robust Q-functions in this context. To address this, we incorporate Q-ensembles in our design to smooth the learned Q-surface, ensuring that the average Q-gradient does not fluctuate excessively. This fluctuation can result from the limited data coverage and the function approximation errors associated with using neural networks. The methods that learns more accurate or smooth Q-functions could be explored to make DAC learn effectively in sparse reward tasks. For instance, increasing the size of the Q-ensemble can lead to a smoother estimation of the Q-gradient. Additionally, employing perturbation regularization—similar to the methods in [1], although [1] focuses on state perturbation while we focus on action perturbation—can aid in learning more robust Q-functions.
>
> [1] RORL: Robust Offline Reinforcement Learning via Conservative Smoothing.

---

> ### Author Response · Authors · 2024-11-21
> **Rebuttal by Authors (3/3)**
>
> >Q5.  Could LCB lead to a systematic bias in the Q-values? How significant is this bias in different task settings?
>
> A5. Thank you for bringing this valuable problem into discussion. The usage of LCB target does introduce a bias that deviates from the Bellman target, which should be the mean value. However, this bias is temporary and primarily affects the early stages of training. During the early stages, the Q-function is under-trained and inaccurate, leading to disagreement of Q-values estimations among the Q-ensembles. This results in a high standard deviation and a low LCB value (can be observed in logs of TensorBoard). Consequently, the value target becomes pessimistic, which is beneficial since we aim to align closely with the behavior policy when the Q-functions are not yet informative. As training progresses, the Q-functions become more accurate and consistent, leading to a decrease in standard deviation and an increase in the LCB value (which can also be observed in TensorBoard). This causes the value target to rise, and the LCB gradually approaches the mean value, aligning with our expectations for a bias-free scenario.
>
> The impact of the LCB varies across different task settings, depending on the inherent randomness in learning Q-functions. For example, we find that using $\rho=1.5$ for Hopper, $\rho=1$ for Walker2D, $\rho=0$ (just mean value) for HalfCheetah in locomotion tasks yields good results. This phenomenon may be explained by the fact that Q-values for HalfCheetah are relatively easy to learn, as most trajectories reach the maximum length without falling. In contrast, Hopper and Walker2D are more prone to falling, which can cause trajectories to end prematurely and complicate the learning of Q-functions in the early stages. Empirically, we find that choosing the smallest $\rho$ for LCB that does not cause divergence of Q-functions yields good results, which makes the tuning of LCB target straightforward. A complete sensitivity analysis with figures can be found in Appendix C.
>
> >Q6. How does the $\eta$ parameter in the objective function influence the optimization process? If gradient vanishing issues arise, particularly when the update frequency of the target policy is high, how should this potential problem be addressed?
>
> A6. In our method, $\eta$ serves as a trade-off between behavior cloning and policy improvement. If $\eta$ is too large, it may overly prioritize behavior policies, resulting in performance that closely resembles the behavior set. Conversely, if $\eta$ is too small, it can lead to policies that stray too far from the behavior support, causing instability in the learning process. We have included a sensitivity analysis of this trade-off in Appendix C.3 and Figure 7(left), except that we conduct an alternative formulation of such trade-off. In the sensitivity analysis, we control $b$ so that $\|\epsilon _ \theta (x _ t , s, t) − \epsilon\|^2 \leq b$, making $\eta$ learnable through dual gradient ascent (line 11, Algorithm 1). Experiments show that both a value of $b$ that is too large or too small can lead to reduced performance. In our revisions, We will revise the main text to help readers easily locate the additional experiments.
>
> Regarding the second question, we can intuitively explain that the Q-gradient will never vanish if the behavior policy is suboptimal, regardless of how often the policy is updated. The update for the target policy consists of both the behavior cloning term and the Q-gradient guidance. If the Q-gradient were to vanish, the update for the target policy would reduce to behavior cloning. Since the behavior policy is suboptimal, the Q-function of the target policy (which is also the Q-function of the behavior policy due to cloning) does not satisfy the Bellman optimality equation. This leads to the existence of state-action pairs that are better than those of the current policy, resulting in a non-vanishing Q-gradient. Therefore, the Q-gradient can only vanish if the behavior policy is optimal or if the behavior cloning term is removed. Since $\eta>0$ is necessary for offline RL, we can safely ensure a policy improvement over the behavior set with non-zero Q-gradients.

---

> > ### Comment · Reviewer_Wqqq · 2024-11-24
> >
> > Thank you for your response. I believe most of my questions have been adequately addressed, so I will keep my score unchanged.

---

> > > ### Author Response · Authors · 2024-11-24
> > > **Thank you for your feedback**
> > >
> > > Thank you very much for recognizing our work. We are also truly grateful for your valuable discussions, which have helped us refine and strengthen our research. We will carefully incorporate the points raised in the rebuttal and discussion into the paper to enhance its clarity and comprehensiveness.

---

### Official Review · Reviewer_GEf6 · 2024-11-04

**Soundness:** 3
**Presentation:** 2
**Contribution:** 2
**Rating:** 6
**Confidence:** 4

**Summary:**

This paper integrates the advantage-weighted regression and diffusion model for offline reinforcement learning. This paper introduces the Diffusion Actor-Critic (DAC) framework, which theoretically formulates KL-constrained policy iteration as a diffusion noise regression problem. The experimental results demonstrate the advantages of DAC in terms of performance and stability. However, several concerns are as follows.

**Strengths:**

Overall the paper is well-written, making readers quickly capture the core idea. There are several advantages of this paper:

1. DAC directly models the target policy as a diffusion model, which is an innovation in the field of offline reinforcement learning.

2. Soft Q guidance adjusts the intensity of Q gradients according to noise scale, encouraging the generation of high-return actions while staying within the behavior support range. This approach effectively prevents actions outside the sampled distribution, enhancing the stability and performance of the algorithm.

**Weaknesses:**

1. The notation of this paper is complicated.

2. The translation between Equations 13 and 14 probably is not acceptable. How could you replace the target policy with the offline dataset? Do you mean behavior policy?

3. The design of Q-ensemble makes the results unconvincing. The Q-ensemble methods can solve the offline RL without any extra regularization. Given the design of the Q-ensemble, it's ambiguous whether the improvement comes from the proposed diffusion policy-constrained iteration or the Q-ensemble. The ablation of Fig. 4 seems to approve this. The baseline methods, such as DQL, should add a Q-ensemble to compare what makes it fairer.

4. Although the paper emphasizes the importance of the LCB objective in stabilizing Q-value estimation, it lacks a deeper analysis of the impact of different pessimism factors, ρ.

5. Insufficient comparison of offline RL, including but not limited to:

[1] RORL: Robust Offline Reinforcement Learning via Conservative Smoothing.

[2] Offline RL with No OOD Actions: In-Sample Learning via Implicit Value Regularization.

Several previous methods using constrained policy learning for offline RL should be discussed at least:

[3] Behavior proximal policy optimization

[4] An Implicit Trust Region Approach to Behavior Regularized Offline Reinforcement Learning

[5] Uni-o4: Unifying online and offline deep reinforcement learning with multi-step on-policy optimization

**Questions:**

See weaknesses.

---

> ### Author Response · Authors · 2024-11-21
> **Rebuttal by Authors (1/2)**
>
> >Q1. The notations are complicated.
>
> A1. Thank you for your valuable feedback. We retain most superscripts and subscripts to ensure that the variables are clearly defined under specific conditions, such as diffusion steps ($t$) and iteration steps ($k$) etc., which may lead to crowded notations. We are sorry that it creates difficulties for readers to follow, and will simplify the notations in our revision to emphasize only the key elements.
>
> >Q2. The concern with the transitions from (13) to (14). How could the expectation be approximated by the behavior dataset?
>
> A2. We apologize for not providing a detailed explanation. We could clarify this concern by regarding the training objectives (13) and (14) as the form of explicit score matching $\mathbb{E} _ {x _ t \sim p(x)} || s _ \theta(x _ t) - \nabla _ {x _ t} \log \pi^* (x _ t)||^2$ over the extended action space $x _ t \in \mathbb{R}^d$. For brevity, we have omitted scalar constants, diffusion steps or states. Ideally, the unbiased estimation (13) would involve expectation using "on-policy" density $\mathbb{E} _ {x _ t \sim \pi^*(x)+ noise}[\cdot]$. However, the target policy $\pi^*$ is not accessible. That's why we approximate the expectation using the "off-policy" density $\mathbb{E} _ {x _ t \sim \pi _ \beta(x)+ noise}[\cdot]$. Additionally, we could not directly sample $x _ 0 \sim \pi _ \beta(x)$ as the behavior policy $\pi _ \beta(x)$ is also unknown. Therefore, we bootstrap from the offline (behavior) data $x _ 0 \sim \mathcal{D}$ to approximate the expectation over the behavior policy, resulting in the surrogate objective (14). An intuitive understanding of this "off-policy" score matching in (14) is that it resembles the commonly used approximated on-policy evaluation in reinforcement learning，except that our method estimates score functions instead of critic values. For example, the approximation of the on-policy visiting distributions in *TRPO* uses old policies (from (2) to (3) in [6]), while the approximation in *Behavior Proximal Policy Optimization* uses the behavior set (from (3) to (4)in [3]).
>
> From this perspective, the transition from (13) to (14) only affects the "sampled locations" used to learn the target score functions. Since the target policy is typically close to the behavior policy in offline RL to mitigate OOD problem, we can safely replace $\mathbb{E} _ {x _ 0 \sim \pi^*(x)}[\cdot]$ with $\mathbb{E} _ {x _ 0 \sim \mathcal{D}}[\cdot]$ ($\approx \mathbb{E} _ {x _ 0 \sim \pi _ \beta(x)}[\cdot]$) without introducing too much bias. Furthermore, this distribution bias can be further reduced by considering the noisy distributions $x _ t \sim \pi^*(x)+ noise$ and $x _ t \sim \pi _ \beta(x)+ noise$, both of which approaches standard normal during the diffusion process.
>
> >Q3. The concern that the improvement may come from the Q-ensemble rather than the proposed diffusion policy-constrained iteration.
>
> A3. We have to emphasize that the improvement of our proposed method over others is not solely based on high scores in the testing environments, but also on the stability of convergence. To demonstrate that the majority of the improvement stems from the proposed soft Q-guidance rather than the Q-ensemble, we have included an ablation study in Figure 3 in our paper. In this study, all designs and parameters are maintained except that soft Q-guidance is replaced with alternatives, such as denoised guidance (similar to DiffusionQL). As seen in Figure 3, the introduction of Q-ensemble does not enhance the stability of convergence for denoised Q-guidance.
>
> Additionally, in Table 1, the reported scores from DiffusionQL utilize *online-model-selection*, which tracks the best models throughout the training process. In contrast, we present the average of the final convergent scores from our method, which demands greater stability from the trained models.
>
> To further validate the effectiveness of DAC and to ensure a fair comparison with DiffusionQL, we conduct additional experiments where we replace the Q-ensemble in DAC with the same number of Qs (num of Qs=2) used in DiffusionQL. We also record the scores for DAC using *online-model-selection* (OMS). It can be seen that, under the same protocol (both using OMS), our method without the Q-ensemble significantly outperforms DiffusionQL in most environments, demonstrating the effectiveness of soft Q-guidance. We also observe that using a Q-ensemble of size 5 or larger yields similar performance.
>
> |Num of Qs in DAC | 2+OMS | 2 | 5 | 10(ours) | 20 |DiffusionQL
> |-|-|-|-|-|-|-
> |hopper-m |103.5|90.5|98.7|101.2|100.9|90.5
> |hopper-m-r|105.3|100.8|103.5|103.1|103.1|101.3
> |hopper-m-e|109.6|93.8|108.2|111.7|111.6|111.1
> |halfcheetah-m|61.0| 58.3|59.3|59.1|59.2|51.1
> |halfcheetah-m-r|57.7|53.0|54.6|55.0|55.4|47.8
> |halfcheetah-m-e|100.8|99.5|99.2|99.1|100.0|96.8
>
> [6] Schulman, John. "Trust Region Policy Optimization." arXiv preprint arXiv:1502.05477 (2015).

---

> ### Author Response · Authors · 2024-11-21
> **Rebuttal by Authors (2/2)**
>
> >Q4. Lacking a deeper analysis of the impact of different pessimism factors $\rho$ for LCB objective.
>
> A4. We apologize for not including the analysis of LCB in the main text due to space constraints, and that we did not mention that a sensitivity analysis of $\rho$ as well as other hyperparameters, such as diffusion steps $T$ and behavior cloning threshold $b$ are in Appendix C. This will be improved in the next version.
>
> For the LCB objective, we experiment with different values of $\rho$, as shown in Figure 7 (right). We use "walker2d-medium-v2" as an example, as we observed that the training curves for other tasks are similar. Our experiments show that a small $\rho$ can lead to an overestimation of Q-functions, causing divergence in the estimated Q-values and resulting in failed learning. Conversely, a larger $\rho$ may underestimate the Q-values, leading to a lower final score. In our experiments, we empirically find that choosing the smallest $\rho \geq 0$ that does not cause divergence in the estimated Q-function is sufficient for achieving good results, making the tuning of $\rho$ quite straightforward.
>
> >Q5 Missing important baselines for comparison or discussion.
>
> A5. Thank you for reminding us of those important papers. In fact, we have already included the comparison with *Offline RL with No OOD Actions: In-Sample Learning via Implicit Value Regularization* (IVR) in Table 1. This paper presents two methods, "Sparse Q-learning" and "Exponential Q-learning" within the framework of the behavior-regularized MDP problem. We present the highest scores attained by both methods in the "IVR" column. As for other papers, we genuinely appreciate you sharing the relevant literature with us and will include the discussion of those as references in our revision.

---

> > ### Comment · Reviewer_GEf6 · 2024-11-23
> > **Response to Authors**
> >
> > I am glad to see the detailed feedback from the authors. The majority of my concerns have been solved, thus I raised the score from 5 to 6 after the rebuttal. A suggestion is to use a different color to highlight the revised content in your revised paper and refer to the detailed paragraphs of the manuscript in the response.

---

> > > ### Author Response · Authors · 2024-11-23
> > > **Thank you for your feedback**
> > >
> > > We truly appreciate your quick reply and constructive feedback! We will definitely enhance the clarity of our paper according to your suggestions. Thank you once again!

---

### Meta-Review · Area_Chair_zWGw · 2024-12-21

**Metareview:**

Diffusion Actor-Critic: Formulating Constrained Policy Iteration as Diffusion Noise Regression for Offline Reinforcement Learning

Summary: The paper introduces Diffusion Actor-Critic (DAC), a new offline reinforcement learning algorithm that utilizes diffusion models to generate target policies. DAC addresses challenges in offline RL by formulating constrained policy iteration as a diffusion noise regression problem, integrating a soft Q-guidance mechanism to ensure the policy remains within behavior support and avoids out-of-distribution (OOD) actions. Unlike prior diffusion-based methods, DAC does not require explicit density estimation, simplifying the training process. Leveraging a Q-ensemble with a lower confidence bound (LCB) for critic learning, DAC enhances stability and prevents overestimation. Experimental evaluations on the D4RL benchmark demonstrate that DAC outperforms existing methods.

Comments: We have received 4 expert reviews, with scores 6, 6, 6,8, and the average score is 6.50.

Reviewers gave positive comments about many aspects of this paper. More than one reviewer commented that  DAC's use of diffusion models to directly model target policies and integrate soft Q-guidance is innovative. Reviewers also commented on the technically grounded approach and analysis.  The experiments are extensive and they show superior performance of DAC. Reviewers appreciated the clear presentation and the insightful discussions.

The reviewers have also provided some comments to strengthen the paper. One major comment is the need for additional comparison with recent works that combine diffusion models and RL. The reviewers have also asked for additional experiments and ablation studies. I am glad to see that the authors have partially addressed these comments during the rebuttal process. I recommend the authors to make the changes to their manuscript according to these comments.

**Additional Comments On Reviewer Discussion:**

Please see the "Comments" in the meta-review.

---

### Decision · Program_Chairs · 2025-01-22

Accept (Poster)